# Retrospective Evaluation of Various Serological Assays and Multiple Parameters for Optimal Diagnosis of Lyme Neuroborreliosis in a Routine Clinical Setting

Tamara van Gorkom,[a,c] Willem Voet,[b] Gijs H. J. van Arkel,[a,c] Michiel Heron,[a] B. J. A. Hoeve-Bakker,[c] Daan W. Notermans,[c] Steven F. T. Thijsen,[a] Kristin Kremer[c]*

aDepartment of Medical Microbiology and Immunology, Diakonessenhuis Hospital, Utrecht, The Netherlands
bDepartment of Neurology, Diakonessenhuis Hospital, Utrecht, The Netherlands
cCentre for Infectious Diseases Research, Diagnostics and Laboratory Surveillance, Centre for Infectious Disease Control, National Institute for Public Health and the Environment (RIVM), Bilthoven, The Netherlands

**ABSTRACT** Laboratory diagnosis of Lyme neuroborreliosis (LNB) is challenging, and validated diagnostic algorithms are lacking. Therefore, this retrospective cross-sectional study aimed to compare the diagnostic performance of seven commercial antibody assays for LNB diagnosis. Random forest (RF) modeling was conducted to investigate whether the diagnostic performance using the antibody assays could be improved by including several routine cerebrospinal fluid (CSF) parameters (i.e., leukocyte count, total protein, blood-CSF barrier functionality, and intrathecal total antibody synthesis), two-tier serology on serum, the CSF level of the B-cell chemokine (C-X-C motif) ligand 13 (CXCL13), and a *Borrelia* species PCR on CSF. In total, 156 patients were included who were classified as definite LNB (*n* = 10), possible LNB (*n* = 7), or non-LNB patient (*n* = 139) according to the criteria of the European Federation of Neurological Societies using a consensus strategy for intrathecal *Borrelia*-specific antibody synthesis. The seven antibody assays showed sensitivities ranging from 47.1% to 100% and specificities ranging from 95.7% to 100%. RF modeling demonstrated that the sensitivities of most antibody assays could be improved by including other parameters to the diagnostic repertoire for diagnosing LNB (range: 94.1% to 100%), although with slightly lower specificities (range: 92.8% to 96.4%). The most important parameters for LNB diagnosis are the detection of intrathecally produced *Borrelia*-specific antibodies, two-tier serology on serum, CSF-CXCL13, Reibergram classification, and pleocytosis. In conclusion, this study shows that LNB diagnosis is best supported using multiparameter analysis. Furthermore, a collaborative prospective study is proposed to investigate if a standardized diagnostic algorithm can be developed for improved LNB diagnosis.

**IMPORTANCE** The diagnosis of LNB is established by clinical symptoms, pleocytosis, and proof of intrathecal synthesis of *Borrelia*-specific antibodies. Laboratory diagnosis of LNB is challenging, and validated diagnostic algorithms are lacking. Therefore, this retrospective cross-sectional study aimed to compare the diagnostic performance of seven commercial antibody assays for LNB diagnosis. Multiparameter analysis was conducted to investigate whether the diagnostic performance using the antibody assays could be improved by including several routine (CSF) parameters. The results of this study show that LNB diagnosis is best supported using the detection of intrathecally produced *Borrelia*-specific antibodies, two-tier serology on serum, CSF-CXCL13, Reibergram classification, and pleocytosis. Furthermore, we propose a collaborative prospective study to investigate the potential role of constructing a diagnostic algorithm using multiparameter analysis for improved LNB diagnosis.

**KEYWORDS** *Borrelia*, Lyme neuroborreliosis, cerebrospinal fluid, intrathecal antibody synthesis, antibody index, two-tier serology, random forest, multiparameter analysis, Reibergram

**Ad Hoc Peer Reviewer** Jean-Luc Murk

Address correspondence to Tamara van Gorkom, tvgorkom@diakhuis.nl.

*Present address: Kristin Kremer, KNCV Tuberculosis foundation, The Hague, the Netherlands.

The authors declare no conflict of interest.

[This article was published on 11 April 2022 with a byline that lacked Daan W. Notermans. The byline was updated in the current version, posted on 25 April 2022.]

Lyme borreliosis (LB), which is caused by *Borrelia burgdorferi sensu lato*, is the most common tick-borne disease in temperate regions of the Northern Hemisphere (1). LB is a multisystem disease, and the most frequent clinical symptom is an expanding skin rash also known as erythema migrans (1, 2). If untreated, the *Borrelia* bacterium can disseminate to other body parts, such as the peripheral and/or central nervous system (Lyme neuroborreliosis [LNB]), joints (Lyme arthritis), or heart (Lyme carditis), or cause acrodermatitis chronica atrophicans (ACA) (3). In Europe and North America, LNB is seen in approximately 3% to 16% of LB cases (4–7) and often presents as a painful meningoradiculitis with or without cranial nerve involvement, although various combinations of other neurological complaints may occur as well (3, 8). In the Netherlands, the annual incidence rate for LNB in 2010 was 2.6 per 100,000 inhabitants, which comprised one third of the total incidence rate of disseminated LB (9).

The diagnosis of LNB must be supported by laboratory tests, because the clinical symptoms of LNB are nonspecific. The European Federation of Neurological Societies (EFNS) recommends the detection of an intrathecal immune response to *B. burgdorferi sensu lato* together with the presence of pleocytosis ($\geq$5 leukocytes/$\mu$L) in the cerebrospinal fluid (CSF) (3). Proof of intrathecally produced *Borrelia*-specific antibodies requires simultaneous measurement of *Borrelia*-specific antibodies in CSF and serum of a CSF-serum pair, which should be interpreted relative to the total amount of antibodies in CSF and serum and taking the blood-CSF barrier functionality into consideration (3, 10). The interpretation of the test results, however, can be complicated, as negative test results do not exclude LNB and positive test results are no indication of active disease. A negative test result in the first few weeks after infection can be explained by the absence of detectable antibody levels, which have to be built up at the start of the infection (11–13). For antibody tests, sensitivities between 55% and 90% have been reported for symptom durations of less than 6 weeks (11, 14–18). As the immune response against *Borrelia* expands over time (19–21), the sensitivity improves as the infection progresses and can ultimately reach 100% (11, 15, 22). Lower sensitivities have also been reported for antibody assays that are based on a single antigen compared to those of antibody assays based on multiple antigens (18, 23, 24). Furthermore, negative test results can be obtained when the antigens present in the assay do not match those of the *B. burgdorferi sensu lato* strain causing disease. This mismatch can be explained by the intra- and interspecies heterogeneity of *B. burgdorferi sensu lato* (25–30) and/or the antigenic variation the bacterium can apply during the course of disease (31). A negative test result can also be caused by antibiotic treatment prior to the lumbar puncture (LP), as this might abrogate the immune response (32, 33). A positive test result can be proof of an active LNB, but can also be the result of a previous, yet cleared, infection as antibody persistence has been reported after successful antibiotic treatment (34, 35).

In clinical practice, proof of intrathecal *Borrelia*-specific antibody synthesis for LNB diagnostics is based on either the detection of these antibodies in CSF-serum pairs and subsequent calculation of a *Borrelia*-specific antibody index (AI) (14–16, 18, 22, 24, 34, 36–38) or the detection of these antibodies in CSF only (39–41). Many commercial assays are available for the detection of *Borrelia*-specific antibodies, and various studies have evaluated the performance of these assays for LNB diagnostics (16, 18, 23, 24, 38, 40). A drawback of most of these studies is that study populations were used that were not representative of the clinical setting in which the antibody assays are used. Therefore, this study aimed to compare the diagnostic performance of seven commercial antibody assays for the diagnosis of LNB using a cross-sectional study design. Furthermore, a random forest (RF) model was constructed for each antibody assay to investigate whether the diagnostic performance found for each assay could be improved by including various routine CSF parameters (i.e., leukocyte count, total protein, blood-CSF barrier functionality, and intrathecal total antibody synthesis). Other parameters added to each RF model included *Borrelia*-specific serum antibodies using

**TABLE 1** Classification of the 156 study participants based on the guidelines of the European Federation of Neurological Societies (EFNS) (3) and consensus strategy

| EFNS criteria and consensus strategy used to classify the 156 study participants (no./total [%]) | | | | Classification of patients[a] | | |
|---|---|---|---|---|---|---|
| Clinical symptoms suggestive of LNB[b] | Pleocytosis (CSF leukocyte count of ≥5 leukocytes/$\mu$L) | Consensus strategy for intrathecal *Borrelia*-specific Ab synthesis[c] | Other cause for symptoms | dLNB | pLNB | non-LNB |
| Yes (56/156 [35.9]) | Yes (17/56 [30.4]) | Yes (10/17 [58.8]) | 0/10 (0.0) | 10 | 0 | 0 |
| | | No (7/17 [41.2]) | 3/7 (42.9)[d] | 0 | 4 | 3 |
| | No (39/56 [69.6]) | Yes (3/39 [7.7]) | 0/3 (0.0) | 0 | 3 | 0 |
| | | No (36/39 [92.3]) | | 0 | 0 | 36 |
| No (100/156 [64.1]) | Yes (19/100 [19.0]) | Yes (0/19 [0.0]) | | 0 | 0 | 0 |
| | | No (19/19 [100]) | | 0 | 0 | 19 |
| | No (81/100 [81.0]) | Yes (0/81 [0.0]) | | 0 | 0 | 0 |
| | | No (81/81 [100]) | | 0 | 0 | 81 |
| Total | | | | 10/156 (6.4) | 7/156 (4.5) | 139/156 (89.1) |

[a]Patients are classified as definite Lyme neuroborreliosis (dLNB), possible LNB (pLNB), or non-LNB patient based on the EFNS guidelines (3) and consensus strategy using the flow chart in Fig. S1.
[b]Clinical symptoms suggestive of Lyme neuroborreliosis (LNB) were assumed to be present when a request for the detection of intrathecal *Borrelia*-specific antibody synthesis was done at our laboratory at the time of active disease in the past. Clinical symptoms suggestive for LNB or an alternative diagnosis that ruled out LNB as well as test results (i.e. pleocytosis and intrathecal *Borrelia*-specific antibody synthesis determined by the CSF-serum assays) needed for patient classification are shown in Table S2. For patients for whom clinical symptoms were not relevant for final classification, a diagnosis was specified in Table S2 only in case of a pathological immunoglobulin (Ig)M and/or IgG antibody index value in at least one of the five CSF-serum assays, and/or a positive test result in at least one of the two CSF-only assays, and/or a positive *Borrelia* species PCR result on CSF, and/or a positive CSF-CXCL13 result.
[c]The consensus strategy entailed that intrathecal *Borrelia*-specific antibody (Ab) synthesis was considered proven only if the majority of the CSF-serum assays under investigation (i.e., IDEIA, Medac ELISA, *recom*Bead assay, Serion ELISA, and Enzygnost ELISA) showed a pathological *Borrelia*-specific IgM and/or IgG AI value (≥1.5).
[d]For three patients, the diagnosis of LNB was ruled out, as another cause for their symptoms was found. One patient was diagnosed with neurosyphilis, one patient had residual complaints due to a previously treated LNB, and one patient had an isolated paralysis of the flexor pollicis due to a Schwannoma in the shoulder, see also Table S2.

two-tier serology, the CSF level of the B-cell chemokine (C-X-C motif) ligand 13 (CXCL13) (42), and a *Borrelia* species PCR on CSF.

## RESULTS

**Study population.** In total, 150 (13.7%) of 1,098 consecutive patients who underwent at least one LP in the predefined study period were included in the current study. Six additional LNB patients were included from outside the predefined study period, all of whom had taken part in two other studies (43, 44). Details on the selection of the 156 patients have already been published, these patients have also been used to evaluate two commercial CSF-CXCL13 assays (45).

**Classification of the study population using the EFNS guidelines and consensus strategy.** All patients were classified as definite LNB, possible LNB, or non-LNB patient based on the EFNS guidelines (3) and consensus strategy according to the flowchart in Fig. S1. Details with regard to clinical symptoms suggestive for LNB, an alternative diagnosis that ruled out LNB, and test results (i.e., pleocytosis and intrathecal *Borrelia*-specific antibody synthesis determined by the CSF-serum assays) needed for the classification of patients are shown in Table S2. The number of possible and definite LNB patients in this study differed slightly from that in our previous study (45), as intrathecal *Borrelia*-specific antibody synthesis was based on either a consensus strategy (this study) or the IDEIA results (previous study).

Ten (6.4%) of the 156 patients were classified as definite LNB patient, and 7 (4.5%) of the 156 patients were classified as possible LNB patient (Table 1). Of the seven possible LNB patients, four (57.1%) had pleocytosis and three (42.9%) had intrathecal *Borrelia*-specific antibody synthesis according to the consensus strategy. A total of 139 (89.1%) of the 156 patients were classified as non-LNB patient. Thirty-nine (28.1%) of them had clinical symptoms suggestive of LNB, of whom 36 (92.3%) had neither pleocytosis nor intrathecal *Borrelia*-specific antibody synthesis according to the consensus strategy and 3 (7.7%) had pleocytosis only. For these three patients, another cause for their symptoms was found (Table S2). One hundred (71.9%) of the 139 patients classified as non-LNB patient did not have clinical symptoms suggestive of LNB. Eighty-one of them had neither pleocytosis nor intrathecal *Borrelia*-specific antibody synthesis according to the consensus strategy, and the remaining 19 had pleocytosis only.

**Demographic characteristics and clinical parameters among the three study groups.** Table 2 shows a detailed overview of the demographic characteristics and clinical parameters among the three study groups. Pleocytosis, an elevated total protein level in CSF, and a positive *Borrelia* species PCR result on CSF was found among definite LNB patients more often than among possible LNB and non-LNB patients, and this was significantly more often than among non-LNB patients. CSF-CXCL13 positivity was found among definite LNB patients significantly more often than among possible LNB and non-LNB patients. No significant differences were observed among the three study groups with regard to gender, age, symptom duration, and CSF-glucose levels.

***Borrelia*-specific IgM and IgG results on serum among the three study groups.** All definite and possible LNB patients had a positive (or equivocal) C6 enzyme-linked immunosorbent assay (ELISA) result on serum, and this was significantly higher than that among non-LNB patients (Table 2). Two-tier serology results on serum showed that the percentage of positive test results for *Borrelia*-specific immunoglobulin (Ig)M, IgG, and the combined IgM and IgG results, hereafter referred to as overall Ig results, was comparable between definite and possible LNB patients. These percentages were significantly higher among those two patient groups than among non-LNB patients, except for the percentage of positive test results for *Borrelia*-specific IgM between definite LNB and non-LNB patients.

**Blood-CSF barrier functionality among the three study groups.** Measurement of CSF and serum albumin concentrations and the subsequent calculation of the CSF/serum quotient for albumin (Q Alb) provided insight into the functionality of the blood-CSF barrier. A dysfunctional blood-CSF barrier was found among definite LNB patients more often than among possible LNB and non-LNB patients, and this was significantly more often than among non-LNB patients (Table 2).

**Intrathecal total IgM and total IgG synthesis among the three study groups.** Intrathecal total antibody synthesis among definite LNB patients was based on an IgM with or without an IgG response (Table S2). Intrathecal total antibody synthesis among possible LNB patients was based on a solitary IgM response and among non-LNB patients based on an IgM and/or IgG response. Intrathecal total IgM synthesis was found among definite and possible LNB patients significantly more often than among non-LNB patients (Table 2). Intrathecal synthesis of total IgG with or without total IgM was more common among definite LNB patients than among possible LNB and non-LNB patients, although this was significant only between definite LNB and non-LNB patients.

***Borrelia*-specific IgM and IgG AI results among the three study groups.** For each of the five CSF-serum assays, the percentage of positive AI results for *Borrelia*-specific IgM, IgG, or both among definite LNB patients was significantly higher than that among non-LNB patients (Table 3). Most of these percentages were also higher among possible LNB than among non-LNB patients, and this was significant for *Borrelia*-specific IgG and overall Ig, except when the IDEIA was used. For *Borrelia*-specific IgM, however, this was significant only using the Enzygnost IgM ELISA.

For all definite LNB patients, and for all possible LNB patients without pleocytosis ($n = 3$), a positive *Borrelia*-specific AI result was based on an IgG response with or without IgM (Table S2). Of the possible LNB patients with pleocytosis ($n = 4$), two had a positive *Borrelia*-specific AI result for IgM only in either one or two CSF-serum assays. Twelve (8.6%) of the 139 non-LNB patients had a positive *Borrelia*-specific AI result for IgM and/or IgG in a minority of the CSF-serum assays (Table S2).

***Borrelia*-specific IgM and IgG in CSF among the three study groups.** All definite and most of the possible LNB patients had a positive C6 ELISA result on CSF, and this C6 ELISA positivity was significantly higher than that among non-LNB patients (Table 3). Using the *recom*Line immunoblot (IB) on CSF, IgM was not detected at all. *Recom*Line IgG IB positivity using the revised interpretation criteria of the manufacturer (Table S3), which were elaborated on throughout this article, was more common among definite LNB patients than among possible LNB and non-LNB patients, although this was significant only between definite LNB and non-LNB patients (Table 3).

**TABLE 2** Detailed overview of the demographic and clinical parameters among definite LNB, possible LNB, and non-LNB patients

| Characteristic[a] | Value for indicated patient group[b] | | | Raw *P* value for BH comparison[b,e] | | |
|---|---|---|---|---|---|---|
| | dLNB (n = 10)[c] | pLNB (n = 7)[d] | non-LNB (n = 139) | dLNB vs pLNB | dLNB vs non-LNB | pLNB vs non-LNB |
| Gender (no. of males [%]) | 7 (70.0) | 5 (71.4) | 66 (47.5) | 1.000 | 0.203 | 0.266 |
| Age (mean [95% CI]/[range]) | 61.2 (48.1–74.3)/(10.7–89.2) | 54.1 (46.1–62.0)/(42.1–74.3) | 51.8 (49.1–54.6)/(17.2–83.4) | 0.133 | 0.063 | 0.740 |
| Duration of symptoms in days (geometric mean [95% CI]/[range])[f] | 26.3 (11.9–58.0)/(3.0–174) | 51.9 (19.1–140)/(8.0–288) | 64.1 (44.3–92.8)/(0.0–2911) | 0.364 | 0.075 | 0.633 |
| Pleocytosis | | | | | | |
| CSF leukocyte count ≥5 leukocytes/µL (no. [%]) | 10 (100) | 4 (57.1) | 22 (15.8) | 0.051 | <0.001[l] | 0.019[k] |
| CSF leukocyte count/µL (geometric mean [95% CI]) | 76.7 (38.9–151)/(8.3–394) | 6.7 (3.2–14.2)/(2.0–21.0) | 1.1 (0.8–1.6)/(0.0–821) | <0.001[l] | <0.001[l] | 0.001[l] |
| Glucose in CSF in mmol/l (geometric mean [95% CI]/[range]) | 3.3 (2.9–3.7)/(2.3–5.1) | 3.8 (3.6–4.1)/(3.5–4.6) | 3.6 (3.5–3.7)/(1.0–7.7) | 0.012[k] | 0.029[k] | 0.127 |
| Total protein in CSF | | | | | | |
| Elevated total protein in CSF (yes [%])[g] | 6 (60.0) | 0 (0.0) | 10 (7.2) | 0.035[k] | <0.001[l] | 1.000 |
| Total protein in g/L (mean [95% CI]/[range]) | 940 (687–1,190)/(430–1,490) | 461 (370–553)/(300–650) | 450 (389–512)/(170–4,280) | 0.006[j] | <0.001[l] | 0.279 |
| Positive CXCL13 result on CSF | 9 (90.0) | 1 (14.3) | 2 (1.4) | 0.004[j] | <0.001[l] | 0.138 |
| Positive *Borrelia* species PCR result on CSF | 2 (20.0) | 0 (0.0) | 0 (0.0) | 0.485 | 0.004[j] | 1.000 |
| C6 ELISA on serum | 10 (100) | 7 (100) | 38 (27.3) | 1.000 | <0.001[l] | <0.001[l] |
| Two-tier serology on serum[h] | | | | | | |
| *Borrelia*-specific IgM (no. [%]) | 3 (30.0) | 3 (42.9) | 5 (3.6) | 0.644 | 0.010[k] | 0.003[j] |
| *Borrelia*-specific IgG (rev)[h] (no. [%]) | 9 (90.0) | 6 (85.7) | 29 (20.9)[h] | 1.000 | <0.001[l] | <0.001[l] |
| *Borrelia*-specific IgG (old)[h] (no. [%]) | | | 27 (19.4) | 1.000 | <0.001[l] | <0.001[l] |
| *Borrelia*-specific IgM and/or IgG (rev)[h] (no. [%]) | 9 (90.0) | 6 (85.7) | 31 (22.3)[h] | 1.000 | <0.001[l] | 0.001[l] |
| *Borrelia*-specific IgM and/or IgG (old)[h] (no. [%]) | | | 29 (20.9) | 1.000 | <0.001[l] | <0.001[l] |
| Albumin | | | | | | |
| Dysfunctional blood-CSF barrier (no. [%]) | 9 (90.0) | 3 (42.9) | 22 (15.8) | 0.101 | <0.001[l] | 0.097 |
| Q albumin (mean × 10$^{-3}$) ([95% CI]/[range]) | 13.6 (9.9–17.2)/(5.7–24.2) | 7.0 (4.7–9.3)/(3.7–12.5) | 6.1 (5.1–7.2)/(1.1–72.3) | 0.019[k] | <0.001[l] | 0.188 |
| Intrathecal total antibody synthesis[i] | | | | | | |
| Intrathecal total IgM (no. [%]) | 7 (70.0) | 3 (42.9) | 7 (5.0) | 0.350 | <0.001[l] | 0.007[l] |
| Intrathecal total IgG (no. [%]) | 5 (50.0) | 0 (0.0) | 9 (6.5) | 0.044[k] | <0.001[l] | 1.000 |
| Intrathecal total IgM and/or IgG (no. [%]) | 7 (70.0) | 3 (42.9) | 14 (10.1) | 0.350 | <0.001[l] | 0.035[k] |

[a] CI, confidence interval; CSF, cerebrospinal fluid; CXCL13, B-cell chemokine (C-X-C motif) ligand 13; Q, quotient.

[b] Patients are categorized as definite Lyme neuroborreliosis (dLNB), possible LNB (pLNB), or non-LNB patient based on the EFNS guidelines (3) and consensus strategy using the flow chart in Fig. S1.

[c] Six (60.0%) of the 10 definite LNB patients were part of the consecutive patients included between August 2013 and June 2016, and 4/10 (40.0%) were selected from outside this period, see also Table S2.

[d] Five (71.4%) of the seven possible LNB patients were part of the consecutive patients included between August 2013 and June 2016, and 2/7 (28.6%) were selected from outside this period, see also Table S2.

[e] BH, Benjamini-Hochberg.

[f] Durations of symptoms for definite and possible LNB patients are also listed in Table S2.

[g] An elevated total protein concentration in the CSF is age dependent (reference range: 150 to 300 mg/mL for ages ≤10 years, 200 to 500 mg/mL for ages between 10 and 40 years, and 250 to 800 mg/mL for ages >40 years [73]).

[h] Two-tier serology on serum was performed using the *recom*Line IgM and IgG immunoblot (IB). The manufacturer of the *recom*Line IB revised the interpretation of the *recom*Line IgG IB in January 2019 by increasing the point value of the VlsE band (Table S3). For two non-LNB patients, the *recom*Line IgG IB result changed from negative to equivocal (equivocal results were scored positive), see also Table S2. Consequently, results are shown that include both the revised (rev) and old interpretation criteria.

[i] Intrathecal total IgM and/or total IgG synthesis is proven if the intrathecal fraction is larger than 10% as described by Reiber (64).

[j] Significant *P* value after applying the Benjamini-Hochberg procedure (FDR ≤ 2.0%).

[k] Nonsignificant *P* value after applying the Benjamini-Hochberg procedure (FDR > 2.0%).

**TABLE 3** Results of the five CSF-serum assays and two CSF-only assays among definite LNB, possible LNB, and non-LNB patients

| Assay | Antibody class | No. of cases with a positive result per total (%) for indicated patient group[a] | | | Raw P value for BH comparison[a,d] | | |
|---|---|---|---|---|---|---|---|
| | | dLNB (n = 10)[b] | pLNB (n = 7)[c] | non-LNB (n = 139) | dLNB vs pLNB | dLNB vs non-LNB | pLNB vs non-LNB |
| IDEIA | IgM | 2/10 (20.0) | 1/7 (14.3) | 0/139 (0.0) | 1.000 | 0.004[i] | 0.048[j] |
| | IgG | 7/10 (70.0) | 1/7 (14.3) | 0/139 (0.0) | 0.050 | <0.001[i] | 0.048[j] |
| | IgM and/or IgG | 7/10 (70.0) | 1/7 (14.3) | 0/139 (0.0) | 0.050 | <0.001[i] | 0.048[j] |
| Medac ELISA | IgM | 4/10 (40.0) | 1/7 (14.3) | 0/139 (0.0) | 0.338 | <0.001[i] | 0.048[j] |
| | IgG | 10/10 (100) | 3/7 (42.9) | 0/139 (0.0) | 0.015[j] | <0.001[i] | <0.001[i] |
| | IgM and/or IgG | 10/10 (100) | 4/7 (57.1) | 0/139 (0.0) | 0.051 | <0.001[i] | <0.001[i] |
| recomBead assay[e] | IgM | 4/10 (40.0) | 0/6 (0.0)[e] | 0/139 (0.0) | 0.234 | <0.001[i] | 1.000 |
| | IgG | 10/10 (100) | 3/6 (50.0)[e] | 4/138 (2.9)[e] | 0.036[j] | <0.001[i] | 0.001[i] |
| | IgM and/or IgG | 10/10 (100) | 3/5 (60.0)[e] | 4/138 (2.9)[e] | 0.095 | <0.001[i] | <0.001[i] |
| Serion ELISA[f] | IgM | 5/9 (55.6)[f] | 1/7 (14.3) | 0/138 (0.0)[f] | 0.145 | <0.001[i] | 0.048[j] |
| | IgG | 9/9 (100)[f] | 3/7 (42.9) | 6/138 (4.3)[f] | 0.019[j] | <0.001[i] | 0.005[i] |
| | IgM and/or IgG | 9/9 (100)[f] | 3/7 (42.9) | 6/138 (4.3)[f] | 0.019[j] | <0.001[i] | 0.005[i] |
| Enzygnost ELISA[g] | IgM | 3/5 (60.0)[g] | 2/5 (40.0)[g] | 1/139 (0.7) | 1.000 | <0.001[i] | 0.003[i] |
| | IgG | 5/5 (100)[g] | 3/5 (60.0)[g] | 3/139 (2.2) | 0.444 | <0.001[i] | <0.001[i] |
| | IgM and/or IgG | 5/5 (100)[g] | 5/5 (100)[g] | 4/139 (2.9) | 1.000 | <0.001[i] | <0.001[i] |
| C6 ELISA | IgM and/or IgG | 10/10 (100) | 6/7 (85.7) | 5/139 (3.6) | 0.412 | <0.001[i] | <0.001[i] |
| recomLine IB[h] | IgM | 0/10 (0.0) | 0/7 (0.0) | 0/139 (0.0) | 1.000 | 1.000 | 1.000 |
| | IgG (rev)[h] | 8/10 (80.0)[h] | 1/7 (14.3)[h] | 5/139 (3.6)[h] | 0.015[j] | <0.001[i] | 0.259 |
| | IgG (old)[h] | 7/10 (70.0)[h] | 0/7 (0.0) | 0/139 (0.0) | 0.010[j] | <0.001[i] | 1.000 |
| | IgM and/or IgG (rev)[h] | 8/10 (80.0)[h] | 1/7 (14.3)[h] | 5/139 (3.6)[h] | 0.015[j] | <0.001[i] | 0.259 |
| | IgM and/or IgG (old)[h] | 7/10 (70.0)[h] | 0/7 (0.0) | 0/139 (0.0) | 0.010[j] | <0.001[i] | 1.000 |

[a]Patients are classified as definite Lyme neuroborreliosis (dLNB), possible LNB (pLNB), or non-LNB patient based on the EFNS criteria (3) and consensus strategy using the flow chart in Fig. S1.
[b]Six (60.0%) out of 10 dLNB patients were part of the consecutive patients included between August 2013 and June 2016, and 4/10 (40.0%) were selected from outside this period, see also Table S2.
[c]Five (71.4%) out of seven pLNB patients were part of the consecutive patients included between August 2013 and June 2016, and 2/7 (28.6%) were selected from outside this period, see also Table S2.
[d]BH, Benjamini-Hochberg.
[e]For three cases, either the IgM AI value (one pLNB patient) or the IgG AI value (one pLNB and one non-LNB patient) could not be determined by the recomBead assay due to insufficient material.
[f]For two cases, one dLNB and one non-LNB patient, the IgM and IgG AI values could not be determined by the Serion ELISA due to insufficient sample material.
[g]For seven cases, five dLNB and two pLNB patients, the IgM and IgG AI values could not be determined by the Enzygnost ELISA, because the ELISA was taken of the market.
[h]The manufacturer of the recomLine immunoblot (IB) revised the interpretation of the recomLine IgG IB in January 2019 by increasing the point value of the VlsE band (Table S3). For seven cases, one dLNB, one pLNB, and five non-LNB patients, the recomLine IgG IB result changed from negative to equivocal (equivocal results were scored positive), see also Table S2. Consequently, results are shown that include both the revised (rev) and old interpretation criteria.
[i]Significant P value after applying the Benjamini-Hochberg procedure (FDR ≤ 2.0%).
[j]Nonsignificant P value after applying the Benjamini-Hochberg procedure (FDR > 2.0%).

**Comparison of *Borrelia*-specific IgM and IgG in CSF and serum.** Detection of *Borrelia*-specific antibodies in CSF is no direct proof that these antibodies are intrathecally produced and can also be explained by passive diffusion from the blood. C6 ELISA positivity in CSF among definite LNB patients did not differ from that in serum (100% [10/10], both) (Table S2). C6 ELISA positivity in CSF among possible LNB patients was comparable to that in serum (85.7% [6/7] and 100% [7/7], respectively) ($P$ = 1.000). Among non-LNB patients, however, C6 ELISA positivity in CSF was significantly lower than that in serum (3.6% [5/139] and 27.3% [38/139], respectively) ($P$ < 0.001; false-discovery rate [FDR] ≤ 2.0%).

The overall Ig *recom*Line IB positivity in CSF among definite LNB patients was comparable to that in serum (80.0% [8/10] and 90% [9/10], respectively) (Table S2) ($P$ = 1.000). Among possible LNB patients, the overall Ig *recom*Line IB positivity in CSF was lower than that in serum (14.3% [1/7] and 85.7% [6/7], respectively) ($P$ = 0.074). Among non-LNB patients, the overall Ig *recom*Line IB positivity in CSF was significantly lower than that in serum (3.6% [5/139] and 25.2% [35/139], respectively) ($P$ < 0.001; FDR ≤ 2.0%).

**Diagnostic performance of the seven antibody assays.** The sensitivities of the IgM assays to diagnose LNB among definite and possible LNB patients ranged from 0.0% (*recom*Line IgM IB) to 50.0% (Enzygnost IgM ELISA) (Fig. 1A). The sensitivities of the IgG assays were higher and ranged from 47.1% (IDEIA IgG) to 81.3% (*recom*Bead

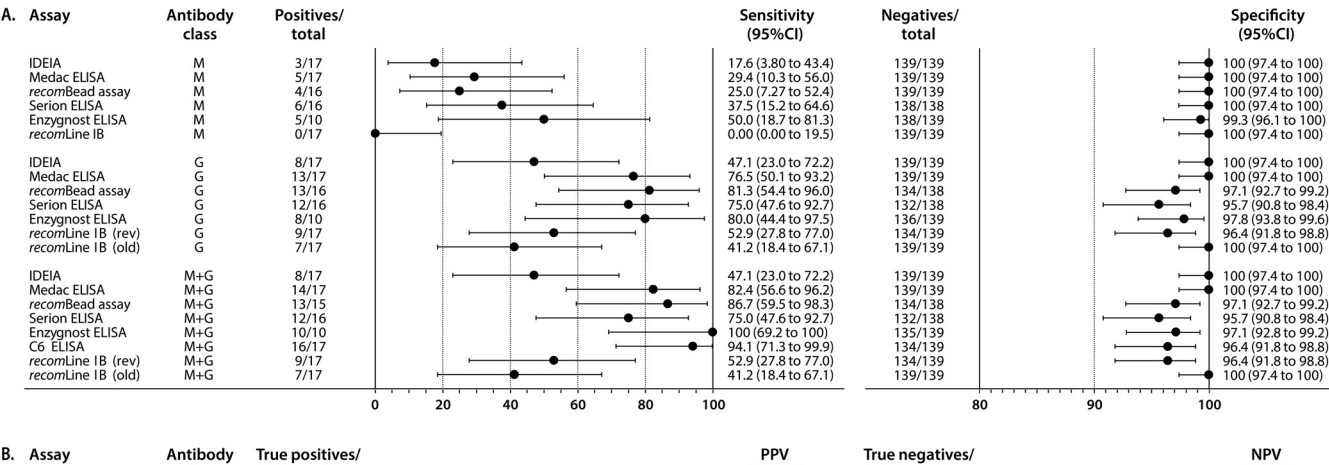

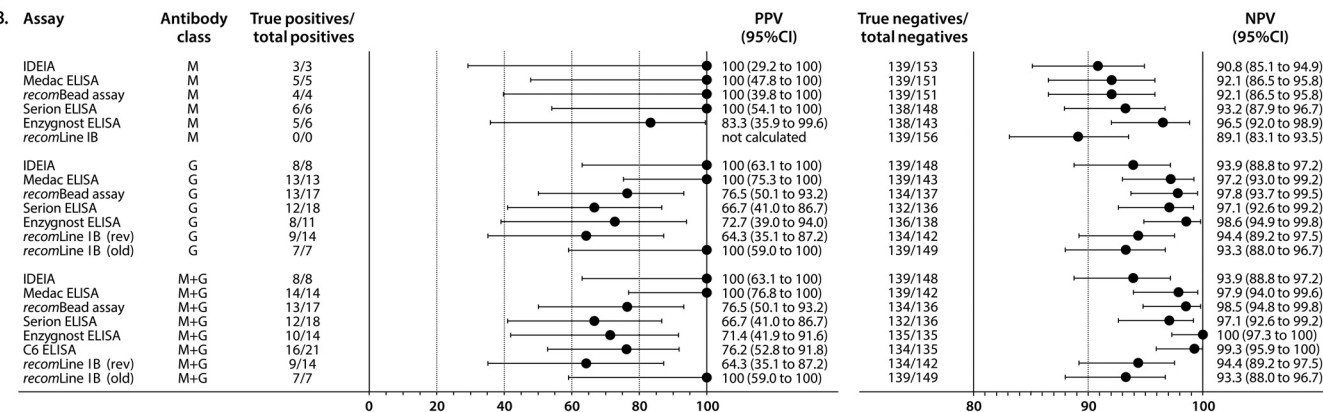

**FIG 1** Overview of the sensitivity and specificity (A) and the positive (PPV) and negative predictive value (NPV) (B) and 95% confidence intervals (CIs) of the five antibody assays tested on cerebrospinal fluid (CSF)-serum pairs and the two antibody assays tested on CSF only for IgM (M), IgG (G), or IgM and IgG combined (M+G). Cases consisted of definite and possible LNB patients, and controls consisted of non-LNB patients. The positives per total (A) are based on the number of pathological AI values (CSF-serum assays) or positive test results (CSF-only assays) among all the cases and are used to calculate the sensitivity. The negatives per total (A) are based on the number of normal AI values (CSF-serum assays) or negative test results (CSF-only assays) among all the controls and are used to calculate the specificity. The true positives (B) are cases that have either a pathological AI value (CSF-serum assays) or a positive test result (CSF-only assays) per total positives (i.e., all patients that have a pathological AI value [CSF-serum assays] or a positive test result [CSF-only assays]). The true negatives (B) are controls that have either a normal AI value (CSF-serum assays) or a negative test result (CSF-only assays) per total negatives (i.e., all patients that have a normal AI value [CSF-serum assays] or a negative test result [CSF-only assays]). The manufacturer of the *recom*Line immunoblot (IB) revised the interpretation of the *recom*Line IgG IB in January 2019 by increasing the point value of the VlsE band (Table S3), which had an effect on the test result (Table 3 and S2). Consequently, results are shown that include both the revised (rev) and old interpretation criteria. For the *recom*Line IgM IB, the PPV could not be calculated as this assay yielded no positive test results.

IgG assay) (Fig. 1A). The overall Ig sensitivities of the antibody assays largely overlapped with those for IgG only and ranged from 47.1% (IDEIA) to 100% (Enzygnost ELISA) (Fig. 1A). The overall Ig sensitivities of two assays were slightly higher than those for IgG only, as a solitary IgM response was found for two possible LNB patients in either one (Medac ELISA) or two (Medac and Enzygnost ELISA) assays (Table S2). The sensitivity of the *recom*Bead assay based on the overall Ig results was also slightly higher than that for IgG only, as for one possible LNB patient the IgM result was missing (Table S2). The specificities of all IgM assays were 100%, except for the Enzygnost IgM ELISA for which the specificity was 99.3% (Fig. 1A). The specificities of the IgG assays ranged from 95.7% (Serion IgG) to 100% (IDEIA IgG and Medac IgG) (Fig. 1A). The specificity of the Enzygnost ELISA based on the overall Ig results was slightly lower than that for IgG only due to a solitary IgM response for one non-LNB patient (Table S2). The sensitivity and specificity of the C6 ELISA were 94.1% and 96.4%, respectively. For IgM, IgG, and overall Ig, the sensitivity and specificity of the seven antibody assays did not differ significantly as the respective 95% confidence intervals (CIs) overlapped (Fig. 1A).

The positive predictive value (PPV) of the Enzygnost IgM ELISA was 83.3%, and the PPVs of all other IgM assays were 100%, except for the *recom*Line IgM IB, for which the PPV could not be calculated due to the absence of positive IgM results (Fig. 1B). For the IgG assays, the PPVs ranged from 64.3% (*recom*Line IgG IB) to 100% (IDEIA and

**TABLE 4** The performance characteristics obtained by constructing random forest models for each antibody assay to predict Lyme neuroborreliosis[a]

| Antibody assay | Value of performance characteristic of antibody assay-specific RF models[b] | | | | | |
|---|---|---|---|---|---|---|
| | AUC | pmc | Sensitivity | Specificity | PPV | NPV |
| IDEIA | 0.973 | 7.1 | 94.1 | 92.8 | 61.5 | 99.2 |
| Medac ELISA | 0.991 | 5.1 | 100 | 94.2 | 68.0 | 100 |
| *recom*Bead assay | 0.993 | 4.6 | 100 | 94.9 | 68.2 | 100 |
| Serion ELISA | 0.986 | 5.2 | 100 | 94.2 | 66.7 | 100 |
| Enzygnost ELISA | 0.986 | 3.4 | 100 | 96.4 | 66.7 | 100 |
| C6 ELISA | 0.987 | 4.5 | 94.1 | 95.7 | 72.7 | 99.3 |
| *recom*Line IB (rev)[c] | 0.970 | 7.1 | 94.1 | 92.8 | 61.5 | 99.2 |
| *recom*Line IB (old)[c] | 0.972 | 7.1 | 94.1 | 92.8 | 61.5 | 99.2 |

[a]AUC, area under the curve; pmc, probability of misclassification; PPV, positive predictive value; NPV, negative predictive value; IB, immunoblot.
[b]Each random forest (RF) model included the following 13 predictor variables: the respective antibody assay, two-tier serology on serum, pleocytosis, CSF-CXCL13, total protein in CSF, *Borrelia* species PCR on CSF, and the seven predictor variables based on one or multiple areas of the Reibergram as shown in Table 6.
[c]The manufacturer of the *recom*Line immunoblot (IB) revised the interpretation of the *recom*Line IgG IB in January 2019 by increasing the point value of the VlsE band (Table S3), which had an effect on the test result (Table 3 and S2) Consequently, results are shown that include both the revised (rev) and old interpretation criteria.

Medac ELISA) (Fig. 1B). The PPVs of the antibody assays based on the overall Ig results were almost comparable to those based on IgG only (Fig. 1B). The negative predictive values (NPVs) of the IgM assays ranged from 89.1% (*recom*Line IgM IB) to 96.5% (Enzygnost IgM ELISA) (Fig. 1B). For the IgG assays, the NPVs ranged from 93.9% (IDEIA IgG) to 98.6% (Enzygnost IgG ELISA) (Fig. 1B). The NPVs of the antibody assays based on the overall Ig results were almost comparable to those based on IgG only (Fig. 1B). The PPV and NPV of the C6 ELISA were 76.2% and 99.3%, respectively. For IgM, IgG, and the overall Ig results, the PPVs and negative predictive values (NPVs) of the seven antibody assays did not differ significantly as the respective 95% CIs overlapped (Fig. 1B).

**Potential role of additional parameters for predicting LNB.** Random forest (RF) modeling was performed to investigate if the diagnostic performance using the antibody assays could be improved by including the results of other parameters. The seven RF models performed comparably well, which was reflected by the areas under the curve (AUCs) that ranged from 0.970 to 0.993 (Table 4). The probability of misclassification of the 156 patients ranged from 3.4% (Enzygnost ELISA RF model) to 7.1% (IDEIA and *recom*Line IB RF models). The sensitivities and NPVs of most RF models were higher than the upper limit of the respective 95% CIs obtained using the results of the antibody assays only, except for the C6 and the Enzygnost ELISA (Table 4 and Fig. 1A and B). In contrast, the specificities and PPVs of most RF models were comparable with those of the antibody assays only, except for the IDEIA and the Medac ELISA, for which the specificities and PPVs obtained using RF modeling were below the lower limit of the respective 95% CIs obtained using the results of the antibody assays only.

For each RF model, the relative importance of the 13 predictor variables was visualized in a heat map (Table 5). Overall, the predictor variables in each RF model ranked comparably. The most important diagnostic parameter in predicting LNB was the antibody assay with a mean rank of 1.7, followed by two-tier serology on serum (mean rank of 2.4) and CSF-CXCL13 (mean rank of 2.6). Of all Reibergram-based predictor variables, a dysfunctional blood-CSF barrier with intrathecal total antibody synthesis (i.e., Reibergram area 3) was most important with a mean rank of 4.1 and preceded pleocytosis (mean rank of 5.1). A dysfunctional blood-CSF barrier in the absence of intrathecal total antibody synthesis (i.e., Reibergram area 2) and the *Borrelia* species PCR on CSF contributed the least (Table 5).

**TABLE 5** Heat maps of the relative contribution of the 13 predictor variables included in the random forest models to investigate their contribution in predicting Lyme neuroborreliosis

| Predictor variables included in the RF model[a] | Relative importance of each predictor variable for each (antibody-assay specific) RF model[b] | | | | | | | | |
|---|---|---|---|---|---|---|---|---|---|
| | IDEIA | Medac ELISA | *recom*Bead assay | Serion ELISA | Enzygnost ELISA | C6 ELISA | *recom*Line IB (rev)[c] | *recom*Line IB (old)[c] | Mean rank of predictor variable[d] |
| Antibody assay | 211 | 354 | 303 | 250 | 250 | 318 | 94 | 174 | 1.7 |
| Two-tier serology on serum[e] | 204 | 173 | 157 | 178 | 111 | 145 | 231 | 217 | 2.4 |
| CSF-CXCL13 | 174 | 141 | 170 | 191 | 88 | 153 | 197 | 179 | 2.6 |
| Reibergram; area 3[f] | 146 | 115 | 131 | 118 | 123 | 115 | 145 | 147 | 4.1 |
| Pleocytosis | 125 | 123 | 91 | 137 | 41 | 139 | 148 | 128 | 5.1 |
| Reibergram; all areas separately[f] | 95 | 74 | 126 | 101 | 85 | 108 | 99 | 99 | 5.6 |
| Reibergram; areas 3 and 4[f] | 81 | 60 | 82 | 64 | 68 | 78 | 89 | 86 | 7.4 |
| Reibergram; area 1[f] | 60 | 55 | 115 | 92 | 61 | 90 | 69 | 65 | 7.6 |
| Reibergram; areas 2 and 3[f] | 63 | 46 | 78 | 72 | 56 | 59 | 62 | 63 | 8.9 |
| Reibergram; area 4[f] | 58 | 40 | 63 | 52 | 39 | 55 | 65 | 64 | 10.1 |
| Total protein in CSF | 43 | 40 | 47 | 18 | -29 | 28 | 71 | 43 | 11.0 |
| Reibergram; area 2[f] | 28 | 10 | 43 | 37 | 32 | 18 | 18 | 27 | 12.1 |
| *Borrelia* PCR on CSF | 27 | 28 | 28 | 33 | 0 | 27 | 27 | 27 | 12.3 |

[a]RF, random forest; CSF, cerebrospinal fluid; CXCL13, B-cell chemokine (C-X-C motif) ligand 13.
[b]The relative importance of each predictor variable was calculated as described by Liaw and Wiener (72).
[c]The manufacturer of the *recom*Line immunoblot (IB) revised the interpretation of the *recom*Line IgG IB in January 2019 by increasing the point value of the VlsE band (Table S3), which had an effect on the test result (Table 3 and S2). Consequently, results are shown that include both the revised (rev) and old interpretation criteria.
[d]For each RF model, the 13 predictor variables were ranked based on their relative contribution from 1 (highest contribution) to 13 (lowest contribution). The mean rank of each predictor variable was calculated using the individual ranks obtained in each of the seven RF models and did not include the RF model of the *recom*Line IgG IB results based on the old interpretation criteria.
[e]Two-tier serology on serum was performed using the C6 ELISA as a screening test, and positive (and equivocal) C6 ELISA results were confirmed using the *recom*Line IgM and IgG IB. The two-tier serology results on serum included the *recom*Line IB results obtained with the revised interpretation criteria of the *recom*Line IgG IB (Table 2 and S2).
[f]For each RF model, the contribution of the Reibergram classification was assessed as described in Table 6.

## DISCUSSION

In this retrospective study, the diagnostic performance of seven antibody assays for the diagnosis of LNB was evaluated. A clinically well-defined study population was used consisting of all consecutive patients from whom CSF and serum were drawn in the routine clinical setting of our hospital and who fulfilled the inclusion criteria. Patients were classified using the EFNS guidelines (3), and intrathecal *Borrelia*-specific antibody synthesis was considered proven using a consensus strategy. RF modeling was performed to assess the utility of additional parameters for predicting LNB.

Of all performance characteristics determined in this study (i.e., sensitivity, specificity, PPV, and NPV), the sensitivity of the seven antibody assays to diagnose LNB among definite and possible LNB patients showed the largest variation (range: 47.1% to 100%), although none of the differences were statistically significant. In general, differences in sensitivity between antibody assays can be influenced by several factors, such as the antigens present in the assay (24, 38, 46). These antigens might be expressed at different time points (20, 21) or might not match the antigens expressed by the *B. burgdorferi sensu lato* strain causing disease due to inter- and intraspecies heterogeneity and/

or antigenic variation (25–31). Overall, it is reasonable to assume that antibody assays based on multiple antigens or whole-cell lysates are expected to give rise to a higher number of positive test results among cases than assays based on a single or a limited number of antigens. Indeed, the sensitivity of the IDEIA, based on a single antigen, was the lowest (i.e., 47.1%) and the sensitivity of the Enzygnost ELISA, based on a whole-cell lysate, was the highest (i.e., 100%). Other studies that investigated multiple antibody assays based on one (i.e., the IDEIA) or multiple antigens also reported the lowest sensitivity for the IDEIA (18, 23, 24).

Besides the large variation in sensitivity between the antibody assays, the sensitivity of most assays did not reach 100%, and this could be explained by the case definition used. In this study, both definite and possible LNB patients were included as cases, which is preferable from a clinical point of view to avoid undertreatment of LNB patients. It was hypothesized previously that possible LNB patients with pleocytosis most likely represent early LNB patients for whom the expanding antibody response is below the detection limit of the antibody assay (47, 48). In this study, this hypothesis is supported by the presence of a solitary *Borrelia*-specific IgM response in two possible LNB patients with pleocytosis, underlining the need for both IgM and IgG testing in LNB diagnostics, as was mentioned before (14, 21). This hypothesis is further supported by a paper from Hansen and Lebech (15), who also reported a low sensitivity for the IDEIA among LNB patients with a short disease duration (sensitivity of 17% for symptoms duration of ≤7 days), which increased to 100% for LNB patients with a disease duration of more than 6 weeks. Early antibiotic treatment can also affect sensitivity since it can abrogate the immune response and, consequently, result in (false) negative test results among cases; however, in this study, antibiotic treatment for LNB had started after the LP was performed (Table S2).

When antibody assays are used that are based on multiple antigens or whole-cell lysates, more (false) positive test results can be expected among controls as well, which leads to a lower specificity. Positive test results among non-LNB patients, which were found mainly for the IgG CSF-serum assays, indeed underline the positive correlation between the number of antigens present in the assay and the number of positive test results. Furthermore, antibody assays based on whole-cell lysates can generate false-positive test results due to the presence of cross-reactive antigens (49). Two non-LNB patients, one with active neurosyphilis and one who had been treated for active neurosyphilis in the past, had a positive *Borrelia*-specific AI result in either the Serion IgG or the Enzygnost IgG ELISA, which could be explained by cross-reactive *Treponema* antibodies (16, 50). As none of the non-LNB patients with a positive *Borrelia*-specific AI result had pleocytosis, except for the patient with active neurosyphilis, an active LNB infection was not likely (47, 48).

Considering that the use of a CSF-serum assay and the calculation of an AI is rather complicated, an assay tested on CSF only would be more convenient and preferable in routine clinical practice. The interpretation of positive test results using a CSF-only assay, however, is complicated because a positive test result can be caused by intrathecal *Borrelia*-specific antibody synthesis or be the result of passive diffusion of these antibodies from the blood or of a traumatic LP. Of the two CSF-only assays tested in this study, the C6 ELISA performed best and might be useful as a screening assay since the NPV was 99.3%. Positive C6 ELISA results on CSF, however, should be confirmed using a CSF-serum assay and subsequent AI calculation that corrects for a dysfunctional blood-CSF barrier to prove intrathecal *Borrelia*-specific antibody synthesis.

In addition to intrathecal *Borrelia*-specific antibody synthesis, the results of other parameters, such as an elevated CSF leucocyte count (3, 14, 28), a dysfunctional blood-CSF barrier (3, 14), intrathecal total antibody synthesis with an IgM dominance (14, 28, 51), and elevated CSF-total protein (3, 14, 28) and CSF-CXCL13 levels (3, 28, 42, 45), can support the diagnosis of LNB. In our study, these findings were confirmed, as all these parameter results were found among definite LNB patients more often than among

non-LNB patients. These findings, thus, strengthen the correct classification of the patients in our study and prompted us to assess the additional value of these parameters in the diagnosis of LNB. RF modeling showed that additional parameters could, indeed, be helpful in the diagnosis of LNB by increasing the sensitivity and NPV, although with a loss in specificity and PPV. In clinical practice, however, overtreating some patients at the cost of not missing true LNB patients is preferred. Overall, two-tier serology on serum, CSF-CXCL13, a dysfunctional blood-CSF barrier with proof of intrathecal total antibody synthesis (Reibergram area 3), and pleocytosis contributed the most to the increased diagnostic performance. To minimize undertreatment, antibody assays with a high NPV are preferred. The EFNS recommends using an AI calculation to prove intrathecal synthesis of *Borrelia*-specific antibodies (3), and the need for this is confirmed in our study. The NPVs of the antibody assays only and those of the RF models showed that RF modeling using a Reiber-based CSF-serum assay is preferred, as the respective NPVs were highest. The results obtained with RF modeling are promising and open up the possibility of defining a diagnostic algorithm for LNB diagnostics.

This study had some limitations. First, some CSF-serum assays lacked results for a few patients, which could have influenced the test performance of these assays. Second, due to the low LNB incidence, few LNB patients were included within the predefined study period. Therefore, six additional LNB patients were included from outside this period. As the total number of LNB patients included in the current study was comparable to the 15 patients expected to be diagnosed with LNB in the predefined study period, we believe that the cross-sectional design of the study holds as has been discussed in more detail previously (45). Third, 20.9% of the non-LNB patients were seropositive for *Borrelia*-specific IgG, whereas the IgG seroprevalence of the Dutch population is 4% to 8% (52). This suggests a selection bias in our study population, although one could argue that a neurologist is more inclined to perform an LP when *Borrelia*-specific antibodies are detected in the blood. This has no consequence for the evaluation of the seven antibody assays, since this reflects routine clinical practice and underlines the need of nonbiased, consecutively selected patient samples for the evaluation of diagnostic assays (53). Fourth, a bias toward older patients was introduced in this study by the inclusion criteria, as at least 1,250 $\mu$L of CSF and 110 $\mu$L of serum had to be present before a patient could be included in order to perform the multiple antibody assays under investigation. In general, less patient material is collected from children than from adults. Indeed, of all the 423 consecutive patients from whom a CSF and serum sample was drawn less than 24 h apart (see Fig. 1 of the previously published manuscript [45]), 61 (14.4%) were children (age <18 years; data not shown). In contrast, of the 150 consecutive patients that had sufficient patient material and were included in this study (see Fig. 1 of the previously published manuscript [45]), only 2 (1.3%) were children (age <18 years; data not shown).

Between the start and publication of this study, some antibody assays that performed well in this study (i.e., Enzygnost, Medac, and C6 ELISA) have been taken off the market. This was partly caused by the new, more stringent quality requirements for *in vitro* diagnostics, which triggered manufacturers to discontinue the production of these assays because of increased costs (54). This development might result in a movement toward the development of monopolies offering diagnostic assays, limiting the diagnostic repertoire (55) and making the availability of assays vulnerable, as was shown recently during the severe acute respiratory syndrome coronavirus 2 pandemic (56).

The main strengths of this study are the cross-sectional design (53) and the well-defined study population. The results obtained in this study should be confirmed, preferably using a prospective design, aiming at including more patients. Because of the relatively low LNB incidence, this is ideally done in an (inter)national joint collaboration using a multiparameter diagnostic algorithm in an effort to standardize LNB diagnostics. Furthermore, this study shows that the Serion ELISA is a suitable assay for the detection of intrathecal *Borrelia*-specific antibody synthesis, which, to our knowledge, has not been reported before.

In conclusion, this study shows that LNB diagnostics is best supported using an approach that includes the detection of intrathecally produced *Borrelia*-specific antibodies using a Reiber-based AI calculation, two-tier serology on serum, CSF-CXCL13, Reibergram classification, and pleocytosis. Furthermore, a collaborative prospective study is proposed to investigate if a standardized diagnostic algorithm can be developed using multiparameter analysis for improved LNB diagnosis.

## MATERIALS AND METHODS

**Study population.** Retrospectively, and regardless of their clinical presentation and age, all consecutive patients were eligible for inclusion if a CSF sample and a blood sample (drawn within 24 h of the LP) had been sent to the microbiology laboratory of the Diakonessenhuis Hospital, Utrecht, the Netherlands in the period between August 2013 and June 2016. Until the start of this study in 2017, leftover material from these patients had been stored at $-20°C$ and/or $-80°C$. Prior to the start of this study, all samples had been freeze-thawed once to aliquot for use in the various assays in this study as well as for another study (45), which ran in parallel. Aliquoted samples were stored at $-20°C$ until use. A prerequisite for patients to be included was the availability of at least 1,250 $\mu$L of CSF and 110 $\mu$L of serum. Patients from whom the CSF was (visually) hemolytic or who had received treatment with intravenous IgG were excluded from the study, as both could interfere with the test results (57, 58). One patient was excluded because of unreliable test results that implied a sample mix-up. As the final number of LNB patients among the included consecutive patients was limited, we included six patients from outside the predefined study period (from February 2011 to July 2013 and from July 2016 to November 2017) that had been diagnosed with LNB in our hospital. These additional LNB patients had taken part in two other studies of our research group for which only adult patients had been included (43, 44). Due to these previous studies, both CSF and serum from the time of diagnosis had been stored. For the CSF and serum of these additional LNB patients, the same inclusion criteria applied as for the CSF and serum of all consecutive patients.

All CSF-serum pairs used in this study were anonymized. Approval of the local ethics committee was not necessary, as the main goal of our study was to compare various antibody assays and assess if additional parameters could improve LNB diagnostics for which leftover material can be used. We did, however, obtain approval from the hospital board. The results of this study are reported following the guidelines for diagnostic accuracy studies (59).

**Classification of the study population.** The EFNS guidelines (3) were used to classify the patients in the current study. Following these guidelines, definite LNB patients should have (i) clinical symptoms suggestive of LNB in the absence of another cause, (ii) CSF pleocytosis ($\geq$5 leukocytes/$\mu$L), and (iii) intrathecal synthesis of *Borrelia*-specific antibodies. Possible LNB patients should have clinical symptoms suggestive of LNB with either pleocytosis or intrathecally produced *Borrelia*-specific antibodies. Clinical symptoms suggestive of LNB (i.e., fulfillment of the first EFNS criterion) were assumed to be present when a request for the detection of intrathecal *Borrelia*-specific antibody synthesis was done at our laboratory at the time of active disease in the past for which the IDEIA LNB IgM and IgG assay (Oxoid, Hampshire, United Kingdom) was used. If the second and/or third criterion (i.e., pleocytosis and intrathecal *Borrelia*-specific antibody synthesis, respectively) was also fulfilled, another cause for the symptoms had to be excluded according to the EFNS guidelines. To minimize the bias in patient classification by using only the IDEIA results, a consensus strategy for proof of intrathecal synthesis of *Borrelia*-specific antibodies was applied. This strategy entailed that intrathecal *Borrelia*-specific antibody synthesis was considered proven only if the majority of the antibody assays tested on CSF-serum pairs in this study showed a pathological *Borrelia*-specific (IgM and/or IgG) AI value (Fig. S1).

***Borrelia*-specific antibody detection in CSF-serum pairs and CSF only.** Seven commercial antibody assays were selected, which were based on different techniques, different *Borrelia* antigens, and different quantification methods (Table S3). For five assays, referred to here as CSF-serum assays, the detection of intrathecally produced *Borrelia*-specific IgM and/or IgG was done by testing the CSF and serum of each patient simultaneously. Results of these CSF-serum assays were used to calculate a *Borrelia*-specific AI value as described below. For two assays, *Borrelia*-specific IgM and/or IgG antibodies were determined in CSF only and are referred to here as CSF-only assays. The seven antibody assays were performed according to the respective manufacturer's instructions, unless otherwise specified here.

The first CSF-serum assay was the second-generation IDEIA LNB assay (Oxoid), referred to here as IDEIA, for the detection of IgM and IgG (Table S3) (44). As the IDEIA was part of the routine LNB diagnostics in our hospital, CSF-serum pairs of patients for whom LNB was suspected had already been tested with this assay at the time of active disease in the past. Consequently, these results were retrieved from the laboratory information system. The CSF-serum pairs of patients for which a request for the detection of intrathecal *Borrelia*-specific antibody synthesis was not done at the time of active disease in the past were tested in batch. CSF-serum pairs were tested only in the singular due to the limited amount of sample material, except CSF-serum pairs for which the CSF sample had an optical density (OD) value of $\geq$0.100, which were repeated. The second CSF-serum assay was the *Borrelia* ELISA Medac (Medac GmbH, Hamburg, Germany), referred to here as Medac ELISA, for the detection of IgM and IgG (Table S3). The third CSF-serum assay was the *recom*Bead *Borrelia* 2.0 multiplex bead-assay (Mikrogen Diagnostik GmbH, Neuried, Germany), referred to here as the *recom*Bead assay, for the detection of IgM and IgG (Table S3). For both the Medac ELISA and the *recom*Bead assay, CSF-serum pairs with an equivocal AI value (1.3 $\leq$ AI $<$ 1.5) should have been repeated following the manufacturers' protocol (Table S3), but

this was not done due to limited sample material. The fourth CSF-serum assay tested was the *Borrelia burgdorferi* SERION ELISA classic (Institute Virion/Serion GmbH, Würzburg, Germany), referred to here as Serion ELISA, for the detection of IgM and IgG (Table S3). The fifth CSF-serum assay was the Enzygnost Borreliosis/IgM ELISA (Siemens Healthcare Diagnostics, Marburg, Germany) and the Enzygnost Lyme link VlsE/IgG ELISA (Siemens Healthcare Diagnostics), referred to here as the Enzygnost ELISA (Table S3). CSF-serum pairs were tested using an adapted protocol we described previously (60), as an edge effect was established for the Enzygnost IgG ELISA following the standard protocol of the manufacturer. This adapted protocol was also used for the Enzygnost IgM ELISA, for which an edge effect was established as well (data not published).

The first CSF-only assay was the C6 ELISA (Immunetics, Boston, MA, USA), which measures total Ig (IgM and IgG) (Table S3) (61). The protocol of the manufacturer describes its use only for serum. This protocol was used to test the CSF using a 1:5 dilution, similar to the CSF dilution used by van Burgel et al. (40). The second CSF-only assay was the *recom*Line IB (Mikrogen GmbH) for the detection of IgM and IgG (Table S3). For practical reasons, we used the manufacturer's protocol for serum to test the CSF. For testing CSF, however, we used a 1:10 dilution for IgM and a 1:20 dilution for IgG (for serum, a 1:51 dilution for IgM and a 1:101 dilution for IgG is recommended). In January 2019, the manufacturer of the *recom*Line IB revised the interpretation of the *recom*Line IgG IB (Table S3). Both the old and revised interpretation criteria were investigated in the current study (62, 63); however, only the revised interpretation criteria were elaborated on throughout the article. For both the C6 ELISA and the *recom*Line IB, equivocal results were interpreted as positive.

All ELISAs were performed on a Dynex DS2 automated ELISA instrument (Dynex Technologies, Chantilly, VA, USA) and analyzed with the DS-Matrix software (Dynex Technologies). The *recom*Bead assay was performed on a Bio-Plex 200 instrument using the Luminex xMAP technology (Bio-Rad Laboratories, Hercules, CA, USA) and the Bio-Plex Manager software version 6.1 (Bio-Rad Laboratories). The *recom*Line IB was performed on an Autoblot 3000 (Medtec Biolab Equipment, Hillsborough, NC, USA). Subsequently, *recom*Line IB strips were scanned and the intensity of the bands was recorded using *recom*Scan software version 3.4 (Mikrogen GmbH).

**Blood-CSF barrier functionality and intrathecal total antibody synthesis.** To investigate the blood-CSF barrier functionality and the intrathecal total IgM and total IgG synthesis, CSF and serum concentrations of albumin, total IgM, and total IgG were determined at the start of this study by nephelometry and used to calculate the CSF/serum quotients for Q Alb, total IgM (Q IgM), and total IgG (Q IgG) as described previously (45). The Q Alb is used to assess the functionality of the blood-CSF barrier as albumin originates from the blood, and a dysfunctional blood-CSF barrier is proven if the Q Alb exceeds the age-dependent Q Alb (10). The Q IgM and Q IgG are used to assess intrathecal total antibody synthesis. If either one or both quotients show a larger increase than the Q Alb and the intrathecal fraction of total IgM and/or total IgG is larger than 10%, then intrathecal total antibody synthesis is proven (64).

**Calculation of the *Borrelia*-specific AI.** For the CSF-serum pairs, either one of two calculation methods was used to calculate the *Borrelia*-specific IgM and IgG AI value (Table S3). For the IDEIA, intrathecal *Borrelia*-specific antibody synthesis was proven if the fraction of *Borrelia*-specific IgM (or IgG) in the CSF as part of the total amount of IgM (or IgG) in the CSF exceeded that of serum, which is expressed by a *Borrelia*-specific IgM (or IgG) AI value (Table S3). Following the manufacturer's protocol, a *Borrelia*-specific AI value of ≥0.3 was considered pathological, and a *Borrelia*-specific AI value of <0.3 (or an OD value of CSF of <0.150) was considered normal. Because the IDEIA is based on a capture ELISA, a correction for a dysfunctional blood-CSF barrier was not necessary (15). For the other four CSF-serum assays, intrathecal *Borrelia*-specific antibody synthesis was proven by the calculation of a *Borrelia*-specific IgM and IgG AI value according to Reiber and Peter (10). These *Borrelia*-specific AI values were calculated by dividing the CSF/serum quotient of *Borrelia*-specific IgM (or IgG) by Q IgM (or Q IgG) by considering the blood-CSF barrier functionality. As the interpretation criteria shown in Table S3 differed slightly between the four CSF-serum assays, the cutoff for intrathecal pathogen-specific (IgM and/or IgG) antibody synthesis as described by Reiber was used (65). Thus, *Borrelia*-specific AI values of ≥1.5 are considered pathological, *Borrelia*-specific AI values between 0.6 and 1.3 are considered normal, and *Borrelia*-specific AI values of <0.5 are not valid. In the current study, *Borrelia*-specific AI values between 1.3 and 1.5 were considered normal. Furthermore, *Borrelia*-specific AI values were used only if these were above the assay-specific lower cutoff (Table S3).

**Borrelia-specific antibody detection in serum.** *Borrelia*-specific antibodies in serum were determined previously using a two-tier protocol in which the C6 ELISA was used as a screening test and equivocal and positive C6 ELISA results were confirmed using the *recom*Line IB (45). The *recom*Line IB was also performed on C6 ELISA negative sera in order to compare the *recom*Line IB results obtained in CSF with those obtained in serum to gain insight into the origin of the *Borrelia*-specific antibodies. Similar to the *recom*Line IgG IB performed on CSF, for serum we also applied the old and revised interpretation criteria of the *recom*Line IgG IB (Table S3) (62, 63); however, only the revised interpretation criteria were elaborated on throughout the article.

**Clinical symptoms and additional parameters.** Results from a number of other parameters, obtained at the time of active disease in the past, were retrieved from the patient and/or laboratory information system. These results included information about clinical symptoms, total protein and glucose concentrations in the CSF, and CSF leukocyte counts. For CSF samples with elevated erythrocyte counts (i.e., ≥1,000 erythrocytes/μL), the CSF leukocyte count was corrected by subtracting 1 leukocyte/μL for each 1,000 erythrocytes/μL according to Reiber and Peter (10). For patients classified as definite or

**TABLE 6** Overview of the seven predictor variables based on one or multiple Reibergram areas that are included in the random forest models

| No. | Predictor variable | Areas of Reibergram | Target of investigation[a] |
|---|---|---|---|
| 1 | Reibergram; overall | Areas 1, 2, 3, and 4 separately | The effect of the overall Reibergram classification |
| 2 | Reibergram; area 1 | Area 1 vs areas 2, 3, and 4 | The effect of any deviation from normal |
| 3 | Reibergram; area 2 | Area 2 vs areas 1, 3, and 4 | The effect of a dysfunctional blood-CSF barrier only |
| 4 | Reibergram; area 3 | Area 3 vs areas 1, 2, and 4 | The effect of a dysfunctional blood-CSF barrier and intrathecal total antibody synthesis[a] |
| 5 | Reibergram; area 4 | Area 4 vs areas 1, 2, and 3 | The effect of intrathecal total antibody synthesis[a] only |
| 6 | Reibergram; areas 2 and 3 | Areas 2 and 3 vs areas 1 and 4 | The effect of a dysfunctional blood-CSF barrier with/without intrathecal total antibody synthesis[a] |
| 7 | Reibergram; areas 3 and 4 | Areas 3 and 4 vs areas 1 and 2 | The effect of intrathecal total antibody synthesis[a] with/without a dysfunctional blood-CSF barrier |

[a]Intrathecal total antibody (IgM and/or IgG) synthesis is proven if the intrathecal fraction is larger than 10% as described by Reiber (64).

possible LNB patient, information regarding symptom duration and antibiotic treatment for LNB was retrieved from the patient information system.

An in-house *Borrelia* species PCR and two CXCL13 assays on CSF had been performed previously, of which the respective methods and results have been published (45). For the current study, the final CSF-CXCL13 result was based on the combined result of the two CXCL13 assays and was considered negative when either one or both assays were negative and positive when both assays were positive.

**Statistical analysis.** For all assays that determined IgM and IgG separately, the overall Ig results were based on a combination of the results of both immunoglobulins: negative when both IgM and IgG were negative and positive when at least one of these was positive. For all statistical analyses, Rstudio (version 1.4.1717, 2009 to 2021) was used (66). We analyzed all data by performing two-group comparisons. The Fisher's exact test for count data was used for unpaired nominal data, the McNemar's chi-squared test with continuity correction was used for paired nominal data, and the exact Wilcoxon-Mann-Whitney test was used for quantitative unpaired data in a 2 by 2 table using the "stats" package (67). Unpaired nominal data in a 2 by 4 table were analyzed with the approximate Monte Carlo resampling $10^6$ Pearson's chi-squared test using the "coin" package (68). Depending on the distribution, either the (geometric) mean value with the 95% CI or the median value and range were shown. Raw *P* values of <0.050 were statistically significant; however, they were interpreted after correction for the multiple statistical analyses in this study, for which the Benjamini-Hochberg procedure (BH) was applied (69). The false-discovery rate (FDR) was set at the level of 2.0% (i.e., less than one false-positive test result was allowed in our list of rejections).

For each antibody assay, the sensitivity, specificity, PPV, and NPV with 95% CIs were calculated using the "epiR" package (70), for which definite and possible LNB patients were used as cases and all non-LNB patients were used as controls.

To investigate if the diagnostic performance using the antibody assays could be improved by including the results of other parameters, RF modeling was performed to predict LNB (71). For each antibody assay, an RF model was built using the "randomForest" package (72) which included 13 predictor variables. The first six predictor variables comprised (i) one of the seven antibody assays (negative/positive), (ii) two-tier serology on serum (negative/positive), (iii) pleocytosis (no/yes), (iv) CSF-CXCL13 (negative/positive), (v) elevated total protein in CSF (no/yes), and (vi) *Borrelia* species PCR on CSF (negative/positive). Blood-CSF barrier functionality and intrathecal total antibody synthesis were also included in the RF models for which the previously published Reibergram classification was used (45). In short, all patients included in this study were classified to one of four of the five Reibergram areas: Reibergram area 1, a normal blood-CSF barrier without intrathecal total IgM and/or IgG synthesis of >10% (*n* = 107/156), Reibergram area 2, a dysfunctional blood-CSF barrier without intrathecal total IgM and/or IgG synthesis of >10% (*n* = 25/156), Reibergram area 3, a dysfunctional blood-CSF barrier with intrathecal total IgM and/or IgG synthesis of >10% (*n* = 9/156), and Reibergram area 4, intrathecal total IgM and/or IgG synthesis of >10% with a normal blood-CSF barrier (*n* = 15/156) (Table S2). Subsequently, seven predictor variables were constructed based on one or multiple Reibergram areas (Table 6).

For all RF models, definite and possible LNB patients were defined as cases and all non-LNB patients were defined as controls. In total, $10^5$ decision trees were built, and for each tree node, three predictor variables were used to split the tree. The predictions obtained in each RF model were used to construct a receiver operating characteristic (ROC) curve, which was subsequently used to calculate the AUC. For each RF model, the optimal cutoff for predicting LNB was calculated using the point on the ROC curve closest to the upper left corner, where both sensitivity and specificity are 100%, and this was determined by the square root of $[(1 - \text{sensitivity})^2 + (1 - \text{specificity})^2]$. Using the optimal cutoff value, the sensitivity, specificity, PPV, and NPV of each RF model were calculated. As the performance characteristics are based on predictions, 95% CIs were not calculated. For each RF model, the probability of misclassification and the relative importance of each predictor variable were calculated as described by Liaw and Wiener (72). For each RF model, the relative importance of the 13 predictor variables was made visible by construction of a heat map. Subsequently, for each RF model, the predictor variables were ranked from 1 (highest relative contribution) to 13 (lowest relative contribution). These ranks were then used to

calculate the mean rank of each predictor variable for the seven RF models to assess the importance of each predictor variable in predicting LNB.

## SUPPLEMENTAL MATERIAL

Supplemental material is available online only.

**SUPPLEMENTAL FILE 1**, PDF file, 0.8 MB.

## ACKNOWLEDGMENTS

We thank the Dutch Ministry of Health, Welfare and Sports for their financial contribution. The funder had no role in the study design, data collection and interpretation, or the decision to submit the work for publication.

We thank José De Sousa Jorge Ferreira from the Department of Statistics, Informatics and Mathematical Modeling of the Dutch National Institute for Public Health and the Environment for support with the statistical analyses and Timo Versijde from Siemens Healthcare, the Netherlands for help with performing nephelometry.

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
