## [Reviewer comments · Microbiology Spectrum]

Microbiology Spectrum

Retrospective evaluation of various serological assays and multiple parameters for optimal diagnosis of Lyme neuroborreliosis in a routine clinical setting

Tamara van Gorkom, Willem Voet, Gijs van Arkel, Michiel Heron, Dienneke Hoeve-Bakker, Steven Thijsen, and Kristin Kremer

Corresponding Author(s): Tamara van Gorkom, Diakonessen hospital Utrecht

Review Timeline:

Submission Date:	January 7, 2022
Editorial Decision:	February 7, 2022
Revision Received:	February 25, 2022
Accepted:	February 25, 2022

Editor: Catherine Brissette

Reviewer(s): Disclosure of reviewer identity is with reference to reviewer comments included in decision letter(s). The following individuals involved in review of your submission have agreed to reveal their identity: Jean-Luc Murk (Reviewer #1); Alje P. van Dam (Reviewer #2)

Transaction Report:

DOI: <https://doi.org/10.1128/spectrum.00061-22>

February 7, 2022

Dr. Tamara van Gorkom
Diakonessen hospital Utrecht
Medical Microbiology and Immunology
Bosboomstraat 1
Utrecht, Utrecht 3582 KE
Netherlands

Re: Spectrum00061-22 (Retrospective evaluation of various serological assays and multiple parameters for optimal diagnosis of Lyme neuroborreliosis in a routine clinical setting)

Dear Dr. Tamara van Gorkom:

While a small study, both the reviewers and I felt this is a very useful contribution to the field. Please address the concerns of the reviewers, which should not require any additional experimentation.

Link Not Available

Sincerely,

Catherine Brissette

Journals Department
Reviewer comments:

Reviewer #1 (Comments for the Author):

In this manuscript Van Gorkum et al describe their evaluation of several serological assays and additional laboratory assays to diagnose Lyme neuroborreliosis. The methods used in this study are sound and the conclusions are worthwhile. It's a bit unfortunate that not more neuroborreliosis cases could be included in this study, because this might have aided with the comparison of the performance of the serological assays.

I have no further questions or suggestions for this manuscript. I think the word 'proof' in line 520 and 536 should be replaced by 'prove'.

Reviewer #2 (Comments for the Author):

The paper is an impressive result of an extensive analysis of paired CSF-serum samples in various assays for Lyme neuroborreliosis. I have a number of comments.

1. In table 4, details on sensitivity and specificity of the various assays are shown, and it is clear that some patients, especially possible LNB, are negative. As stated by the authors (l.83-88) this could be due to short duration of disease or early treatment. However, this is not discussed in the text. Adding duration of disease -and, if known, early antibiotic treatment to supplementary figure 3 (only relevant for cases, not for controls) and discussing them in the text on sensitivity of antibody assays would greatly add to the value of the manuscript.

2. L. 326: Details on the selection of the 156 patients have already been published (31). The authors refer to reference 31, and one gets the impression that the same panel of patients is described. However, a close look at the number of dLNB, pLNB and non-Lyme cases in this reference show that these differ between the numbers in this manuscript and the numbers in reference 31. The authors should explain these differences. Apparently a few different samples have been used. Alternatively, patients could have been reclassified, but this might have consequences for the analysis in reference 31.

3. The authors state that all patients were eligible if a paired blood and serum sample for Borrelia testing were received, and in addition (L.119-120) that a prerequisite for patients to be included was the availability of at least 1250 µl of CSF and 110 µl of serum. This could well have influenced the results, especially with regard to inclusion of children, from whom usually less CSF is obtained. The authors do not mention anything about this issue. Table 1 suggests that there were few, if any, children included. If pediatric LNB was excluded beforehand, this could be stated. Otherwise, some epidemiological data regarding LNB cases diagnosed in the study period, but excluded for lack of availability of a blood sample or insufficient volume of CSF would be helpful whether these patients differed from the 17 cases included in the study.

Minor comments:

4. PCR for Borrelia species was done, but details (reference?) could be mentioned.

5. To obtain a good impression of a quantitative parameter such as age, duration of disease, pleocytosis, glucose and total protein, ranges or interquartile ranges provide more insight than 95% confidence interval. Since the number of patients with LNB is low, interquartile range would not add much and ranges should be provided. This might be done in a supplementary table if the authors want to keep the 95% CIs in table 3 because of statistics.

6. L. 454-459: The sensitivities and NPVs of most RF models were higher than the upper limit of the respective 95% CIs obtained using the results of the antibody assays only, except for the C6 and the Enzygnost ELISA (Table 5 and Figure 1A/B). In contrast, the specificities and PPVs of most RF models were comparable with those of the antibody assays only, except for the IDEIA and the Medac ELISA for which the specificities and PPVs obtained using RF modelling were below the lower limit of the respective 95% CIs obtained using the results of the antibody assays only. I would conclude that the consequence would be that in case of a negative result in an antibody assay, other parameters should be taken into account to increase sensitivity; in contrast, if an antibody assay is positive, other parameters should not be taken that much into account (at least not as done in the RF model), since this might decrease specificity and PPV. Is this correct? If so, could this be stated?

7. A request for a Lyme test on CSF does not tell that the patient really had clinical symptoms suggestive of LNB as defined by ENFS. Apparently, this was not checked by the authors, and for convenience reasons it was supposed that if a clinician had made a request for a Lyme test on CSF, these symptoms existed. I would consider this acceptable, but in the various passages where this is addressed (methods, table 3 note c, supplementary figure 1) the text should rather be "Clinical symptoms suggestive of LNB were supposed to be present when a request....."

8. L.534-536: The EFNS recommends to use an AI calculation to prove (prove?) intrathecal synthesis of Borrelia-specific antibodies (3) and this is confirmed in our study. I would omit that conclusion, since no analysis was performed on assays on CSF only -except C6 and blot, for which the conclusion does not hold for C6 (see below).

9. L.537-539: The NPVs of the antibody assays only and those of the RF models showed that a Reiber-based CSF-serum assay is preferred as the respective NPVs were highest. This is not correct, the NPV of the C6-assay is 99.3% (figure 1) and therefore higher than most of the Reiber-based assays.

10. Abstract l. 37-38: RF modelling demonstrated that most of the sensitivities of the antibody assays could be improved by including other parameters. The authors probably mean "sensitivity of most antibody assays...."

11. From l.242-249 it can be concluded that "two tier-serology" means C6 EIA in serum followed by immunoblot. However, in table 3 and especially in table 6 -where CSF-serum pairs are tested with other ELISAs- a footnote could be added to clarify this. In clinical practice, if an ELISA would be used to determine whether intrathecal antibodies are present, the same assay would probably be used to screen serum for anti-Borrelia antibodies, but an analysis in which the different ELISAs were considered as screening assays was not done in this study.

Staff Comments:

Preparing Revision Guidelines

Please return the manuscript within 60 days; if you cannot complete the modification within this time period, please contact me. If you do not wish to modify the manuscript and prefer to submit it to another journal, please notify me of your decision immediately so that the manuscript may be formally withdrawn from consideration by Microbiology Spectrum.

Response to the reviewers' comments

General comment to the editor from the author:

Thank you for giving us the opportunity to submit a revised version of the manuscript 'Retrospective evaluation of various serological assays and multiple parameters for optimal diagnosis of Lyme neuroborreliosis in a routine clinical setting' for publication in 'Microbiology Spectrum'. We appreciate the time and effort that you and the reviewers dedicated to providing feedback on our manuscript and are grateful for the insightful comments. We have incorporated most of the suggestions and believe these have improved our manuscript. Please see below our point-by-point response (in italic and blue text) to the reviewers' comments. All line and page numbers refer to the revised manuscript with tracked changes.

Reviewer #1: (Comments for the Author):

In this manuscript Van Gorkum et al describe their evaluation of several serological assays and additional laboratory assays to diagnose Lyme neuroborreliosis. The methods used in this study are sound and the conclusions are worthwhile. It's a bit unfortunate that not more neuroborreliosis cases could be included in this study, because this might have aided with the comparison of the performance of the serological assays.

Comment author: We thank the reviewer for thoroughly reading this manuscript. Indeed, the number of Lyme neuroborreliosis (LNB) patients in our study is limited, but this is inherent to the cross-sectional study design. A case-control study would have been easier to perform and would have included more LNB patients, but has a risk of introducing bias, since such a study excludes patients who are difficult to diagnose (1). By choosing a cross-sectional study design, which comprised almost three years, the number of LNB patients was in line with the expected number of LNB patients to be included in our hospital during the predefined study period, as was discussed on page 28 (lines 592-597).

I have no further questions or suggestions for this manuscript. I think the word 'proof' in line 520 and 536 should be replaced by 'prove'.

Comment author:

We thank the reviewer for bringing this to our attention and the suggested corrections have been made (page 27; lines 567 and 583).

Reviewer #2 (Comments for the Author):

The paper is an impressive result of an extensive analysis of paired CSF-serum samples in various assays for Lyme neuroborreliosis. I have a number of comments.

Comment author:

We would like to thank the reviewer for thoroughly reading this manuscript and for the thoughtful comments and constructive suggestions and have revised accordingly. We feel that the manuscript is greatly improved as a result.

1. In table 4, details on sensitivity and specificity of the various assays are shown, and it is clear that some patients, especially possible LNB, are negative. As stated by the authors (1.83-88) this could be due to short duration of disease or early treatment. However, this is not discussed in the text. Adding duration of disease -and, if known, early antibiotic treatment to supplementary figure 3 (only relevant for cases, not for controls) and discussing them in the text on sensitivity of antibody assays would greatly add to the value of the manuscript.

Comment author:

*We thank the reviewer for bringing this to our attention. In this study, the sensitivities of the antibody assays to diagnose LNB among definite and possible LNB patients ranged between 47.1% and 100% and this might be caused by a number of reasons. As was mentioned in the Introduction section, a negative test result might be explained by a short disease duration or early antibiotic treatment (page 5, lines 83-88 of first submission). However, it might also be caused by other reasons such as the antigen composition of the antibody assays, which might - at least in part - explain the variety in sensitivities found between the assays. Negative test results can also be obtained when the antigens present in the assay do not match the antigens expressed by the *B. burgdorferi* sensu lato strain causing disease. This mismatch can be caused by the intra- and interspecies heterogeneity of *B. burgdorferi* sensu lato (2-7) and/or the antigenic variation that bacterium can apply during the course of disease (8). This has now been added to the Introduction section, including references (page 5, lines 91-94) and to the Discussion section (page 24, lines 508-510). We also adjusted some of the text in the Introduction section covering this issue by adding the following sentences:*

- *page 4-5, lines 83-84: as negative results do not exclude LNB and positive results are no indication of active disease*
- *page 5, lines 87-88: As the immune response against *Borrelia* expands over time (9-11), the.....*

As we had already stated in the Discussion section (page 24, lines 510-513), we assume that a positive correlation might also exist between the number of antigens present in the assay and the number of positive results, both among cases and controls. We have now added to the Introduction section that negative test results can be obtained if a limited number of antigens are used in an antibody assay (page 5, lines 89-90) and elaborate a bit more on this issue in the Discussion section by adding the following sentences:

- *page 24-25, lines 513-517: Indeed, the sensitivity of the IDEIA, based on a single antigen, was the lowest (i.e. 47.1%) and the sensitivity of the Enzygnost ELISA, based on a whole cell lysate, was the highest (i.e. 100%). Other studies investigated multiple antibody assays based on one (i.e. the IDEIA) or multiple antigens also reported the lowest sensitivity for the IDEIA (12-14).*

*As for the symptom duration, we did not observe significant differences between the three groups as was mentioned in the Results section (page 18, lines 375-376). The relatively small number of patients in the study precluded a more detailed analysis on this issue. We do, however, discuss the negative results among possible LNB patients and believe these results support the previous hypothesis stated by others that possible LNB patients with pleocytosis could represent early LNB patients for whom the expanding antibody response is below the detection limit of the antibody assay (15-17). We also state that this hypothesis is supported by the presence of a solitary *Borrelia*-specific IgM response in two possible LNB patients with pleocytosis (page 25, lines 525-526). This hypothesis is further supported by a paper*

from Hansen et al. (18), who reported a sensitivity of the IDEIA of 17% for patients with a disease duration ≤ 7 days, which increased to 100% for patients with a disease duration of more than 6 weeks. This has now been added to the Discussion section (page 25, lines 528-531). As suggested by the reviewer, we have added information regarding the symptom duration among definite and possible LNB patients to Table S3 (new column, and footnote 'g'). We have also added a footnote (f) to Table 3 to refer to Table S3 for information about the duration of symptoms among definite and possible LNB patients.

- *Additional remark with regard to lines 519-528 (page 25): these lines have been relocated when compared to the first submitted version of this manuscript (page 25, lines 502-510), as this text also covers the sensitivity of the antibody assays].*

As for early antibiotic treatment which might abrogate the immune response, we believe this did not play a role in our study as antibiotic treatment had started after the lumbar puncture (LP) was performed. We have now added this to the Discussion section (page 25, lines 531-534). As suggested by the reviewer, we have also the following information about antibiotic treatment to Table S3 (new column, and footnote 'h'):

- *Antibiotic treatment started after the LP, with a median of 0 days [range 0-18]. In total, 15 of the 17 patients that were classified as definite or possible LNB patient had been treated for LNB according to the Dutch guidelines for LB (19). Nine definite LNB patients were treated with ceftriaxone (2 g/day) intravenously for either 14 (n=6; ctrx 14) or 28 (n = 3; ctrx 28) days. One definite LNB patient had started with intravenous ceftriaxone (2 g/day), but after 5 days switched to oral doxycycline (100 mg twice a day) for 25 days because of an allergic reaction (ctrx/doxy). Four of the possible LNB patients received ceftriaxone (2 g/day) intravenously for either 14 (n=3; ctrx 14) or 28 (n = 1; ctrx 28) days. The remaining possible LNB patient received oral doxycycline (100 mg twice a day) from the start for 30 days (doxy 30).*
- *Additional information (not added to the manuscript): Two of the 17 patients, both classified as possible LNB patients, had not been treated for LNB in our hospital at the time of the LP as they had not been diagnosed with active LNB. These two patients neither had pleocytosis nor intrathecally produced Borrelia-specific antibodies detected by the IDEIA, which was used for LNB diagnostics in our hospital at that time. The first patient had noticed a redness on the skin of the leg after a four-day walk six months prior to the LP. At that time, the patient was treated with oral doxycycline, which was prolonged to seven weeks due to non-specific symptoms. These non-specific symptoms included cognitive complaints, motor weakness and sensory disturbance. Six months later, this patient visited the neurology department of our hospital, and magnetic resonance imaging showed microvascular white matter*

lesions. Following the inclusion of this patient in the current study, intrathecal antibodies were detected by the majority (4/5) of the CSF-serum assays (except the IDEIA) and was consequently classified as possible LNB patient. In the absence of pleocytosis and a positive CSF-CXCL13 result, an active infection for this patient seemed unlikely. However, this patient might have had LNB six months earlier for which this patient had been adequately treated at that time, and the intrathecally produced Borrelia-specific antibodies were still detectable at the time of the LP. White matter lesions and the presence of intrathecal Borrelia-specific antibody production has been reported before (20, 21).

The second patient had a radicular syndrome and a foot drop, and had fallen down the stairs 10 days before. Based on the radicular syndrome, an LP was performed, but no elevated CSF leucocyte count was found and the IDEIA performed at that time did not show intrathecal Borrelia-specific antibody synthesis. As symptoms remained, an EMG was performed six weeks later, which showed a compression neuropathy of the peroneal nerve on both sides explaining the symptoms. The positive AI results obtained by the majority (4/5) of the CSF-serum assays(except the IDEIA) might reflect a serological scar from a previous infection or might be false positive. Still, LNB can also not be ruled out even though peripheral neuropathy is mainly associated with ACA (22, 23).

For both disease duration and antibiotic treatment, we have added to the Materials and Methods section that this information was retrieved from the patient information system for definite and possible LNB patients (page 13, lines 271-272).

2. L. 326: Details on the selection of the 156 patients have already been published (31). The authors refer to reference 31, and one gets the impression that the same panel of patients is described. However, a close look at the number of dLNB, pLNB and non-Lyme cases in this reference show that these differ between the numbers in this manuscript and the numbers in reference 31. The authors should explain these differences. Apparently a few different samples have been used. Alternatively, patients could have been reclassified, but this might have consequences for the analysis in reference 31.

Comment author:

We thank the reviewer for pointing out this unclarity. In the published manuscript (17), in which two commercial CXCL13 assays were validated for use in LNB diagnostics, the classification of the study population was done using the EFNS guidelines (24). Following these guidelines, definite LNB patients should have: (i) clinical symptoms suggestive of LNB in the absence of another cause, (ii) CSF pleocytosis (≥ 5 leucocytes/ μ l), and (iii) intrathecal synthesis of Borrelia-specific antibodies. Possible LNB patients should have clinical symptoms suggestive of LNB with either pleocytosis or intrathecally produced Borrelia-specific antibodies. In all other cases, the patient should be classified as a non-LNB patient. In the published manuscript (17), the detection of intrathecally produced Borrelia-specific

IgM and IgG was done using the second generation IDEIA LNB assay (Oxoid, Basingstoke, United Kingdom). In the current study, however, a consensus strategy was used to prove intrathecal synthesis of Borrelia-specific antibodies to minimize the bias in the classification of patients by only using the IDEIA results. Consequently, three LNB patients who were classified as possible LNB patient in the published manuscript (17) (i.e. presence of pleocytosis and absence of intrathecal Borrelia-specific antibody synthesis using the IDEIA), were classified as definite LNB patient in the current study using the consensus strategy (i.e. the majority of the antibody assays tested on CSF-serum pairs showed a pathological Borrelia-specific (IgM and/or IgG) AI value [see Table S3, last three rows of the definite LNB patients]). Two patients who were classified as non-LNB patient in the published manuscript (17) (i.e. absence of pleocytosis and absence of intrathecal Borrelia-specific antibody synthesis using the IDEIA), were classified as possible LNB patient in the current study using the consensus strategy (see Table S3, last two rows of the possible LNB patients). We have now added the following text to the Results section:

- *page 17, lines 531-354: The number of possible and definite LNB patients in this study slightly differed from our previous study (17) as intrathecal Borrelia-specific antibody synthesis was based on either a consensus strategy (this study) or on the IDEIA results (previous study).*

We believe that the different classification of these patients does not have an effect on the inferences made with regard to the usefulness of CSF-CXCL13 in LNB diagnostics. Furthermore, in the discussion of the published manuscript (17), we have elaborated on the three possible LNB patients who had pleocytosis and a negative IDEIA result and hypothesized that these negative IDEIA results are caused by the lower sensitivity of the IDEIA in the early stages of LNB, as has been reported by others (12, 18). We also mention that intrathecal Borrelia-specific antibody synthesis was shown using other CSF-serum assays and refer to the current manuscript. The hypothesis of an early LNB is further strengthened by the IgM and IgG Reibergrams, as these show a disturbed blood-CSF barrier and intrathecal total-IgM synthesis in the absence of intrathecal total-IgG synthesis (17).

3. The authors state that all patients were eligible if a paired blood and serum sample for Borrelia testing were received, and in addition (L.119-120) that a prerequisite for patients to be included was the availability of at least 1250 µl of CSF and 110 µl of serum. This could well have influenced the results, especially with regard to inclusion of children, from whom usually less CSF is obtained. The authors do not mention anything about this issue. Table 1 suggests that there were few, if any, children included. If pediatric LNB was excluded beforehand, this could be stated. Otherwise, some epidemiological data regarding LNB cases diagnosed in the study period, but excluded for lack of availability of a blood sample or insufficient volume of CSF would be helpful whether these patients differed from the 17 cases included in the study.

Comment author:

We understand the concern of the reviewer. In this study, no selection criteria were applied based on age for the consecutively selected patients. This has now been more clearly stated in the Materials and Methods section (page 7, line 119). The six additional LNB patients who were included in this study had taken part in two other studies of our research group (25, 26) and due to the age criterion (≥ 18 years) used in these two studies, these six LNB patients were adult patients. For the CSF and serum of these additional LNB patients, the same inclusion criteria applied as for the CSF and serum of all consecutive patients. This has now been more clearly stated in the Materials and Methods section (page 7, line 134-138).

However, to address the concern of the reviewer, we also investigated the ages among all consecutive patients for whom a CSF and blood sample was drawn less than 24 hours apart, which involved 423 of the 1098 patients (see Figure 1 of our published manuscript (17)). Of these 423 patients, 61 (14.4%) were children. Subsequently, 273 were excluded of whom the majority (269/273 [98.5%]) had insufficient sample material (CSF and/or serum). Of the remaining 150 patients that were finally included in the study, 2 (1.3%) were children. As this is much less than the percentage of children for whom a CSF and blood sample was drawn less than 24 hours apart (i.e. 14.4%). We now mention in the Discussion section that the prerequisite of at least 1250 μl of CSF and 110 μl of serum is a limitation of this study and might have led to a bias towards older patients (page 28-29, lines 603-611), since less sample material (CSF and/or blood) is collected from children.

Minor comments:

4. PCR for *Borrelia* species was done, but details (reference?) could be mentioned.

Comment author:

*We thank the reviewer for pointing out this unclarity. In the Materials and Methods section we refer to the in-house *Borrelia* species PCR and the two CXCL13 assays on CSF. Both methods have been described in detail in the published manuscript (17). We have adjusted part of the text to better emphasize this (page 13-14; lines 274-276).*

5. To obtain a good impression of a quantitative parameter such as age, duration of disease, pleocytosis, glucose and total protein, ranges or interquartile ranges provide more insight than 95% confidence interval. Since the number of patients with LNB is low, interquartile range would not add much and ranges should be provided. This might be done in a supplementary table if the authors want to keep the 95% Cis in table 3 because of statistics.

Comment author:

*We thank the reviewer for this valuable comment and have added the ranges of age, duration of symptoms, pleocytosis (i.e. the CSF leucocyte count/ μl), glucose, total protein, and *Q* albumin to Table 3.*

6. L 454-459: The sensitivities and NPVs of most RF models were higher than the upper limit of the respective 95% CIs obtained using the results of the antibody assays only, except for the C6 and the Enzygnost ELISA (Table 5 and Figure 1A/B). In contrast, the specificities and PPVs of most RF models were comparable with those of the antibody assays only, except for the IDEIA and the Medac ELISA for which the specificities and PPVs obtained using RF modelling were below the lower limit of the respective 95% CIs obtained using the results of the antibody assays only. I would conclude that the consequence would be that in case of a negative result in an antibody assay, other parameters should be taken into account to increase sensitivity; in contrast, if an antibody assay is positive, other parameters should not be taken that much into account (at least not as done in the RF model), since this might decrease specificity and PPV. Is this correct? If so, could this be stated?

Comment author:

We appreciate the reviewer's feedback and we understand the remark. Indeed, it is interesting to hypothesize on this issue. In general, RF modeling is not hindered from including parameters that do not contribute to the prediction. As a positive antibody assay result is no proof of an active infection, other parameters are needed to discriminate a serological scar from active disease. Various CSF parameters can be helpful such as elevated CSF cell counts, a disturbed blood-CSF barrier and/or elevated CSF-CXCL13 levels and should also be considered when LNB is suspected. It would be interesting to see if alternative modeling and/or additional parameters could contribute to the diagnosis of active LNB, especially among cases that have Borrelia-specific antibodies detected intrathecally. We hope to engage on a prospective multicenter effort to address these issues.

7. A request for a Lyme test on CSF does not tell that the patient really had clinical symptoms suggestive of LNB as defined by ENFS. Apparently, this was not checked by the authors, and for convenience reasons it was supposed that if a clinician had made a request for a Lyme test on CSF, these symptoms existed. I would consider this acceptable, but in the various passages where this is addressed (methods, table 3 note c, supplementary figure 1) the text should rather be "Clinical symptoms suggestive of LNB were supposed to be present when a request....."

Comment author:

We thank the reviewer for pointing this out. We have adjusted the text where appropriate (Methods section [page 8, line 151-153], Table 2 [footnote 'c'], supplemental figure S1 [footnote 'a'], and supplemental table S3 [footnote 'b']).

8. L.534-536: The EFNS recommends to use an AI calculation to proof (prove?) intrathecal synthesis of Borrelia-specific antibodies (3) and this is confirmed in our study. I would omit that conclusion, since no analysis was performed on assays on CSF only -except C6 and blot, for which the conclusion does not hold for C6 (see below).

Comment author:

We appreciate the reviewer's feedback. Prompted by this feedback, we adjusted part of the text to better reflect our conclusion and now state that a Reiber-based CSF-serum assay is recommended using RF modeling (page 28, line 585). See also our response to comment #9 of this reviewer.

9. L.537-539: The NPVs of the antibody assays only and those of the RF models showed that a Reiber-based CSF-serum assay is preferred as the respective NPVs were highest. This is not correct, the NPV of the C6-assay is 99.3% (figure 1) and therefore higher than most of the Reiber-based assays.

Comment author:

We thank the reviewer for pointing out this unclarity. The highest NPVs were found using RF modeling and a Reiber-based CSF-serum assay. We changed the text accordingly (page 28, line 585). See also our response to comment #8 of this reviewer.

10. Abstract l 37-38: RF modelling demonstrated that most of the sensitivities of the antibody assays could be improved by include ng other parameters. The 4 authors probably mean "sensitivity of most antibody assays...."

Comment author:

We thank the reviewer for bringing this to our attention. The text has been revised as suggested (page 2; lines 37-38).

11. From l.242-249 it can be concluded that "two tier-serology" means C6 EIA in serum followed by immunoblot. However, in table 3 and especially in table 6 -where CSF-serum pairs are tested with other ELISAs- a footnote could be added to clarify this. In clinical practice, if an ELISA would be used to determine whether intrathecal antibodies are present, the same assay would probably be used to screen serum for anti-Borrelia antibodies, but an analysis in which the different ELISAs were considered as screening assays was not done in this study.

Comment author:

Thank you for this suggestion. The footnote in Table 3 (footnote 'h') has been revised as suggested and a footnote (e) has been added to Table 6 as suggested.

References.

1. Leeflang MM, Ang CW, Berkhout J, Bijlmer HA, Van Bortel W, Brandenburg AH, Van Burgel ND, Van Dam AP, Dessau RB, Fingerle V, Hovius JW, Jaulhac B, Meijer B, Van Pelt W, Schellekens JF, Spijker R, Stelma FF, Stanek G, Verduyn-Lunel F, Zeller H, Sprong H. 2016. The diagnostic accuracy of serological tests for Lyme borreliosis in Europe: a systematic review and meta-analysis. BMC Infect Dis 16:140.

2. Roessler D, Hauser U, Wilske B. 1997. Heterogeneity of BmpA (P39) among European isolates of *Borrelia burgdorferi* sensu lato and influence of interspecies variability on serodiagnosis. *Journal of clinical microbiology* 35:2752-2758.
3. Wang G, Van Dam AP, Schwartz I, Dankert J. 1999. Molecular typing of *Borrelia burgdorferi* sensu lato: taxonomic, epidemiological, and clinical implications. *Clinical microbiology reviews* 12:633-653.
4. Ornstein K, Berglund J, Bergstrom S, Norrby R, Barbour AG. 2002. Three major Lyme *Borrelia* genospecies (*Borrelia burgdorferi* sensu stricto, *B. afzelii* and *B. garinii*) identified by PCR in cerebrospinal fluid from patients with neuroborreliosis in Sweden. *Scand J Infect Dis* 34:341-6.
5. Hansen K, Crone C, Kristoferitsch W. 2013. Lyme neuroborreliosis. *Handb Clin Neurol* 115:559-75.
6. Margos G, Vollmer SA, Ogden NH, Fish D. 2011. Population genetics, taxonomy, phylogeny and evolution of *Borrelia burgdorferi* sensu lato. *Infection, Genetics and Evolution* 11:1545-1563.
7. Brisson D, Baxamura N, Schwartz I, Wormser GP. 2011. Biodiversity of *Borrelia burgdorferi* strains in tissues of Lyme disease patients. *PloS one* 6:e22926.
8. Zhang JR, Hardham JM, Barbour AG, Norris SJ. 1997. Antigenic variation in Lyme disease borreliae by promiscuous recombination of VMP-like sequence cassettes. *Cell* 89:275-85.
9. Craft JE, Fischer D, Shimamoto G, Steere A. 1986. Antigens of *Borrelia burgdorferi* recognized during Lyme disease. Appearance of a new immunoglobulin M response and expansion of the immunoglobulin G response late in the illness. *The Journal of clinical investigation* 78:934-939.
10. Arumugam S, Nayak S, Williams T, di Santa Maria FS, Guedes MS, Chaves RC, Linder V, Marques AR, Horn EJ, Wong SJ. 2019. A multiplexed serologic test for diagnosis of Lyme disease for point-of-care use. *Journal of clinical microbiology* 57:e01142-19.
11. Zajkowska J, Lelental N, Kulakowska A, Mroczko B, Pancewicz S, Bucki R, Kornhuber J, Lewczuk P. 2015. Application of multiplexing technology to the analysis of the intrathecally released immunoglobulins against *B. burgdorferi* antigens in neuroborreliosis. *Immunol Lett* 168:58-63.
12. Cerar T, Ogrinc K, Strle F, Ruzic-Sabljić E. 2010. Humoral immune responses in patients with Lyme neuroborreliosis. *Clin Vaccine Immunol* 17:645-50.
13. Wutte N, Archelos J, Crowe BA, Zenz W, Daghofer E, Fazekas F, Aberer E. 2014. Laboratory diagnosis of Lyme neuroborreliosis is influenced by the test used: comparison of two ELISAs, immunoblot and CXCL13 testing. *J Neurol Sci* 347:96-103.
14. Henningsson AJ, Christiansson M, Tjernberg I, Lofgren S, Matussek A. 2014. Laboratory diagnosis of Lyme neuroborreliosis: a comparison of three CSF anti-*Borrelia* antibody assays. *Eur J Clin Microbiol Infect Dis* 33:797-803.
15. Picha D, Moravcova L, Smiskova D. 2016. Prospective study on the chemokine CXCL13 in neuroborreliosis and other aseptic neuroinfections. *J Neurol Sci* 368:214-20.
16. Henningsson AJ, Gyllemark P, Lager M, Skogman BH, Tjernberg I. 2016. Evaluation of two assays for CXCL13 analysis in cerebrospinal fluid for laboratory diagnosis of Lyme neuroborreliosis. *APMIS* 124:985-990.
17. van Gorkom T, van Arkel GHJ, Heron M, Voet W, Thijsen SFT, Kremer K. 2021. The Usefulness of Two CXCL13 Assays on Cerebrospinal Fluid for the Diagnosis of Lyme Neuroborreliosis: a Retrospective Study in a Routine Clinical Setting. *J Clin Microbiol* 59:e0025521.

18. Hansen K, Lebech AM. 1991. Lyme neuroborreliosis: a new sensitive diagnostic assay for intrathecal synthesis of *Borrelia burgdorferi*-specific immunoglobulin G, A, and M. *Ann Neurol* 30:197-205.
19. CBO. Kwaliteitsinstituut voor de Gezondheidszorg (CBO). Dutch guidelines Lyme disease 2013. <http://www.tekenbeetziekten.nl/wp-content/uploads/2014/08/CBO-richtlijn-Lymeziekte-versie-2013.pdf>. Retrieved 22 July 2013.
20. Agarwal R, Sze G. 2009. Neuro-lyme disease: MR imaging findings. *Radiology* 253:167-73.
21. Logigian EL, Kaplan RF, Steere AC. 1990. Chronic neurologic manifestations of Lyme disease. *N Engl J Med* 323:1438-44.
22. Kristoferitsch W, Sluga E, Graf M, Partsch H, Neumann R, Stanek G, Budka H. 1988. Neuropathy associated with acrodermatitis chronica atrophicans. Clinical and morphological features. *Ann N Y Acad Sci* 539:35-45.
23. Mygland A, Ljostad U, Fingerle V, Rupprecht T, Schmutzhard E, Steiner I, European Federation of Neurological Societies. 2010. EFNS guidelines on the diagnosis and management of European Lyme neuroborreliosis. *Eur J Neurol* 17:8-16, e1-4.
24. Mygland A, Ljostad U, Fingerle V, Rupprecht T, Schmutzhard E, Steiner I. 2010. EFNS guidelines on the diagnosis and management of European Lyme neuroborreliosis. *Eur J Neurol* 17:8-4.
25. van Gorkom T, Sankatsing SUC, Voet W, Ismail DM, Muilwijk RH, Salomons M, Vlaminckx BJM, Bossink AWJ, Notermans DW, Bouwman JJM, Kremer K, Thijsen SFT. 2018. An Enzyme-Linked Immunosorbent Spot Assay Measuring *Borrelia burgdorferi* B31-Specific Interferon Gamma-Secreting T Cells Cannot Discriminate Active Lyme Neuroborreliosis from Past Lyme Borreliosis: a Prospective Study in the Netherlands. *J Clin Microbiol* 56.
26. van Gorkom T, Voet W, Sankatsing SUC, Nijhuis CDM, Ter Haak E, Kremer K, Thijsen SFT. 2020. Prospective comparison of two enzyme-linked immunosorbent spot assays for the diagnosis of Lyme neuroborreliosis. *Clin Exp Immunol* 199:337-356.

February 25, 2022

Dr. Tamara van Gorkom
Diakonessen hospital Utrecht
Medical Microbiology and Immunology
Bosboomstraat 1
Utrecht, Utrecht 3582 KE
Netherlands

Re: Spectrum00061-22R1 (Retrospective evaluation of various serological assays and multiple parameters for optimal diagnosis of Lyme neuroborreliosis in a routine clinical setting)

Dear Dr. Tamara van Gorkom:

The authors have thoroughly addressed the concerns of the reviewers.

Your manuscript has been accepted, and I am forwarding it to the ASM Journals Department for publication. You will be notified when your proofs are ready to be viewed.

Sincerely,

Catherine Brissette
Editor, Microbiology Spectrum
